# Hidden Heterogeneity: When to Choose Similarity-Based Calibration

**Kiri L. Wagstaff**  *kiri.wagstaff@oregonstate.edu*
*School of Electrical Engineering and Computer Science*
*Oregon State University*

**Thomas G. Dietterich**  *tgd@cs.orst.edu*
*School of Electrical Engineering and Computer Science*
*Oregon State University*

**Reviewed on OpenReview:** *https://openreview.net/forum?id=RAOTDqt3hC*

## Abstract

Trustworthy classifiers are essential to the adoption of machine learning predictions in many real-world settings. The predicted probability of possible outcomes can inform high-stakes decision making, particularly when assessing the expected value of alternative decisions or the risk of bad outcomes. These decisions require well-calibrated probabilities, not just the correct prediction of the most likely class. Black-box classifier calibration methods can *improve the reliability* of a classifier's output without requiring retraining. However, these methods are unable to detect subpopulations where calibration could also *improve prediction accuracy*. Such subpopulations are said to exhibit "hidden heterogeneity" (HH), because the original classifier did not detect them. This paper proposes a quantitative measure for HH. It also introduces two similarity-weighted calibration methods that can address HH by adapting locally to each test item: SWC weights the calibration set by similarity to the test item, and SWC-HH explicitly incorporates hidden heterogeneity to filter the calibration set. Experiments show that the improvements in calibration achieved by similarity-based calibration methods correlate with the amount of HH present and, given sufficient calibration data, generally exceed calibration achieved by global methods. HH can therefore serve as a useful diagnostic tool for identifying when local calibration methods would be beneficial.

## 1 Introduction

How do we know when to trust a prediction? Let $f(X)$ be a classifier that outputs a discrete probability distribution $\hat{P}(Y|X)$ over the $K$ possible class labels $\{1, \ldots, K\}$. Ideally, each prediction made by the classifier will be *point-wise calibrated*, that is, the true class distribution for each $X$ matches $\hat{P}$: $P(Y|X) = \hat{P}(Y|X)$. Many investigators have studied a weaker requirement that each distinct output value $g$ is calibrated, such that $P(Y|g) = E[Y]$ over the set of predictions for which $\hat{P}(Y|X) = g$ (Vaicenavicius et al., 2019; Widmann et al., 2019). This ensures aggregate calibration for the set, but it allows individual predictions to have $P(Y|X) \neq \hat{P}(Y|X)$. Others use the same approach but require calibration only for the most-likely class's predicted probability (e.g., Guo et al., 2017; Patel et al., 2021; Luo et al., 2022) or marginal class probabilities (e.g., Zadrozny & Elkan, 2002; Kumar et al., 2019), both of which also do not enforce point-wise calibration. Point-wise calibration of each individual prediction, for its full class distribution, is important for safety-critical applications. Well-calibrated predictions improve the trustworthiness of systems and support downstream cost-sensitive decisions (e.g., medical diagnosis, autonomous driving, financial decisions). Likewise, calibration is necessary when combining or comparing predictions from different sources (Bella et al., 2013) or in classifier cascades that use a low-cost but less accurate classifier's output to decide whether to apply a higher-cost but more accurate secondary classifier (Enomoto & Eda, 2021). Good calibration is

beneficial in any decision making setting in which uncertainty matters (e.g., active learning or classification with a rejection or abstention option).

We focus on an increasingly common use case in which we would like to apply a pre-trained, possibly proprietary, model $\mathcal{M}$ to our own data set $\mathcal{D}$ with corresponding distribution $P_D(Y|X)$. In this scenario, the original training data set is unavailable and $P(Y|X)$, the distribution for which $\mathcal{M}$ was trained (and possibly calibrated), is unknown. Any domain shift between $P$ and $P_D$ could prevent $\mathcal{M}$ from generating reliable predictions on $\mathcal{D}$. Moreover, even in the absence of domain shift, $\mathcal{M}$ may perform poorly on $D$ due to "hidden heterogeneity", which occurs when $\mathcal{M}$ assigns the same posterior probability to items with different true probabilities.

Concerns about poorly calibrated classifiers are not new (e.g., Zadrozny & Elkan, 2001; Niculescu-Mizil & Caruana, 2005), and several post-training calibration correction methods have been developed (e.g., Guo et al., 2017; Kumar et al., 2019; Kull et al., 2019; Alexandari et al., 2020). In general, these methods devise a *calibration map* $\Phi$ that transforms the original predicted probabilities into values that are better calibrated. We denote the output of a classifier $f(x_i)$ applied to item $x_i$ as the probability vector $\hat{p}_i$ of length $K$ (number of classes) that sums to 1 (i.e., resides in the simplex $\Delta_{K-1}$). The calibration map $\Phi : \Delta_{K-1} \mapsto \Delta_{K-1}$ is derived from an independent calibration set $\mathcal{C}$ to transform $\hat{p}_i$ to a more reliable $\hat{q}_i = \Phi(\hat{p}_i)$.

A key limitation of these calibration maps is the implicit assumption that all items with the same predicted probability vector $\hat{p}$ should be given the same correction. Such maps cannot accommodate hidden heterogeneity, which manifests as subpopulations with distinct $P(Y|X)$ values to which the classifier has erroneously assigned the same $\hat{p}$ value. For example, in predicting cancer risk, there could be many different reasons (age, lifestyle, family medical history, etc.) that a given individual is predicted to have $\hat{p} = 0.9$. Even if the predictions satisfy aggregate calibration, this probability could be an over-estimate for some individuals, while for others it could be an under-estimate. No global calibration map can address this heterogeneity to achieve point-wise calibration, because they map all items with the same $\hat{p}$ to the same $\hat{q}$.

We propose a method to quantify hidden heterogeneity (HH) as a signal for when global calibration may be inadequate. Once HH is detected, we face a choice of either (a) training a new classifier on the available calibration data or (b) improving the existing classifier using the calibration data. Because HH is a local phenomenon, a natural way to improve the classifier is to apply a local, similarity-based calibration technique. We introduce two local calibration methods that leverage the location of $x_i$ in feature space to yield $\hat{q}_i = \Phi(\hat{p}_i|x_i)$. These methods determine the calibrated probability $\hat{q}_i$ by taking a weighted vote of data points in the calibration set $\mathcal{C}$. The first method, Similarity-Weighted Calibration (SWC), assigns weights to every point in the calibration set based on similarity to $x_i$. The second method, SWC-HH, uses only items within a local neighborhood defined by the estimated HH. We refer to the weighted number of calibration data points as the "calibration support" for $x_i$, which indicates how much calibration data is available for estimating $\hat{q}_i$. This measure of the calibration quality of $\hat{q}_i$ for each $x_i$ is a unique advantage of local calibration.

We note that any post-hoc calibration method can be viewed as a form of model stacking (Wolpert, 1992), in which the output of the original classifier is transformed via $\Phi$, a model itself. Our SWC and SWC-HH methods are stacking methods that focus on improving local calibration. As a consequence, they also reduce or eliminate HH and can thereby improve classifier accuracy.

The major contributions of this paper are

1. The identification of hidden heterogeneity as a property of classifier predictions that thwarts global calibration methods (Section 3),

2. A method for quantifying hidden heterogeneity to indicate when local calibration is needed (Section 3),

3. Two local calibration methods based on Similarity-Weighted Calibration (Section 4), and

4. Results of experiments that assess the relationship between hidden heterogeneity and calibration, yielding useful guidance for practitioners (Section 5).

We provide context from previous work in Section 2. Key conclusions and limitations of local, similarity-based calibration are discussed in Section 6.

## 2 Related Work

There are several methods for improving the reliability (calibration) of classifier predictions. Many recent advances were inspired by the recognition that deep neural networks in particular may sacrifice calibration to achieve higher generalization accuracy (Guo et al., 2017). Strategies include using calibration-sensitive training methods, if the original training set is available (e.g., via modifications to the loss function, as proposed by Kumar et al., 2018; Mukhoti et al., 2020; Enomoto & Eda, 2021; Tomani & Buettner, 2021), using domain-specific representations that lead to improved calibration (Kalmady et al., 2021), or adopting network architectures that do not use convolutions (Minderer et al., 2021). In contrast, post-hoc calibration correction methods that directly modify the classifier's predictions on new observations, without re-training, can be employed even when the training data (or model) are proprietary or when the data distribution has changed and we wish to recalibrate an existing classifier to extend its applicability.

Global, parametric calibration methods re-map the predicted probabilities, $\hat{p}$, output by the classifier by fitting a chosen functional form (e.g., logistic curve) from the probabilities to the labels to compute $\hat{q} = \Phi(\hat{p})$. For binary classifiers, Platt scaling (Platt, 1999) transforms $\hat{p}_i$ into a value between 0 and 1 using a sigmoid function with two parameters, $A$ and $B$: $\hat{q}_i = \frac{1}{1+e^{A\hat{p}_i+B}}$. The parameters $A$ and $B$ are chosen to optimize the negative log-likelihood of predictions made on the calibration set. Platt scaling was generalized to multi-class problems for neural networks (Guo et al., 2017) via a method called temperature scaling, which operates on the logits $z_i$ (not the probabilities) by optimizing a temperature parameter $T$ in $u_i[k] = e^{z_i[k]/T}$, where $z_i[k]$ is the logit for item $i$ and class $k$, and $u_i[k]$ is the corresponding unnormalized probability. These values are normalized as $\hat{q}_i[k] = \frac{u_i[k]}{\sum_j u_j[k]}$. The same $T$ value is used for all classes. Bias-Corrected Temperature Scaling (Alexandari et al., 2020) adds a bias term for each class.

There are also several approaches that construct probability bins and assign the average (or other aggregate) accuracy within bin $B_b$ as its calibrated probability, $\hat{q}_i := Acc_b, \forall i \in B_b$. Histogram binning (Zadrozny & Elkan, 2001) assigns items to bins based on their uncalibrated predictions $\hat{p}_i$, often using equally-spaced bin boundaries or divided so that each bin has the same number of items ("equal frequency") from the calibration set. Kumar et al. (2019) found that the latter strategy, as well as using a larger number of bins, yields better results. Isotonic regression (Zadrozny & Elkan, 2002) adds further flexibility by optimizing the bin boundaries to minimize the squared loss between $\hat{q}_i$ and $y_i$. Recently, Patel et al. (2021) proposed selecting the bin boundaries to maximize the mutual information between bin predictions $\hat{q}_i$ and $y_i$.

To date, very few calibration methods have leveraged the location of items in feature space, $\mathcal{X}$. Zhao et al. (2020) introduced "individual" (per-item) calibration for regression problems and confidence intervals. Partial specialization for classification problems can be achieved by estimating a different $T$ per subpopulation (unlabeled cluster (Gong et al., 2021) or labeled "domain" (Yu et al., 2022)), then employing linear regression to estimate a new $T'$ for each test item. Our approach operates at a finer (per-item) granularity and is not restricted to probability rescaling. Like our method, the LoRe calibration method (Luo et al., 2022) considers the similarity of the calibration set items to the test item $x$. However, LoRe restricts the similarity calculation to calibration items that fall into a probability bin based on the probability $\max_k \hat{p}_i[k]$ of the highest-probability class. This can produce high variance estimates when the bin contains few calibration items. Our method avoids this problem by considering the full predicted distribution $\hat{p}_i$ when computing similarity. LoRe also only calibrates the highest-probability prediction; it does not produce a calibrated probability distribution over all $K$ classes. Consequently, it does not support downstream tasks such as computing the expected costs of misclassification (in cost-sensitive problems) or re-estimating class probabilities (Alexandari et al., 2020).

One calibration approach that employs similarity to compute the complete $\hat{q}_i[k]$ vector is Similarity-Binning Averaging or SBA-10 (Bella et al., 2009), which creates bins (neighborhoods) that contain an item's 10 nearest neighbors (in Euclidean distance) in an "augmented" feature space $\mathcal{X}^+ = \mathcal{X} \times \Delta_{K-1}$ defined by the item's feature vector $x_i$ of dimension $d$ concatenated with its probability vector $\hat{p}_i \in \Delta_{K-1}$. SBA-10 computes the calibrated probability $\hat{q}_i[k]$ as the probability of class $k$ (in the calibration set) within item $i$'s assigned bin, with each item contributing equally (Bella et al., 2009). In contrast, our approach uses a similarity-weighted contribution from every item in the calibration set, not just the 10 nearest neighbors.

---

**Algorithm 1** Hidden Heterogeneity (HH)

---

**Input**: Test item $x_t$, calibration data $\mathcal{C}$, predicted probabilities $\hat{p}$, and radius $r$
**Output**: Hidden heterogeneity in neighborhood around $x_t$

 1: Construct probability neighborhood around $x_t$: $\mathcal{U}_t = \{x_i \in \mathcal{C} | D_H(\hat{p}_t, \hat{p}_i) < r\}$ (using Eqn. 2).
 2: Train $g_t$ using labeled data in $\mathcal{U}_t$.
 3: Collect model predictions for the neighborhood: $f(\mathcal{U}_t) = \{\hat{p}_i | x_i \in \mathcal{U}_t\}$.
 4: Collect $g_t$ predictions for the neighborhood: $g_t(\mathcal{U}_t) = \{g_t(x_i) | x_i \in \mathcal{U}_t\}$.
 5: Collect labels for the neighborhood: $Y_{\mathcal{U}_t} = \{y_i | x_i \in \mathcal{U}_t\}$.
 6: Calculate $HH_{\mathcal{U}_t}$ using $f(\mathcal{U}_t)$, $g_t(\mathcal{U}_t)$, and $Y_{\mathcal{U}_t}$ (Eqn. 3).

---

## 3 Hidden Heterogeneity

Global post-hoc calibration methods, such as Platt scaling and temperature scaling, perform very well for some data sets and algorithms and less well for others. Similarly, local methods like SBA-10 do not always improve upon these global methods. What causes the failure of global methods, and under what conditions can local methods do better? Our hypothesis is that global post-hoc calibration, which focuses on aggregate calibration, fails when the data exhibits *hidden heterogeneity* (HH) with respect to the predicted probabilities $\hat{p}$. HH characterizes situations where there are subpopulations in the feature space $\mathcal{X}$ to which the classifier assigns the same $\hat{p}$ but that require different calibration corrections.

### 3.1 Hidden Heterogeneity

**Definition 1** A classifier $f$ exhibits *hidden heterogeneity* with respect to a feature space $\mathcal{X}$ if there exists a subregion $\mathcal{U} \subseteq \mathcal{X}$ such that $f(x) \approx \hat{p}$ for all $x \in \mathcal{U}$ and yet $\mathcal{U}$ can be partitioned into $M$ disjoint subregions $\mathcal{U} = \mathcal{U}_1 \coprod \cdots \coprod \mathcal{U}_M$ such that the true class probabilities $P(y|x \in \mathcal{U}_m) \neq P(y|x \in \mathcal{U}_{m'})$ for all distinct pairs $m, m' \in \{1, \ldots, M\}, m \neq m'$.

An extreme example of HH occurs for a classifier that ignores all features and predicts the majority class for all items. Imagine a data set composed of 60% cats and 40% birds, for which a classifier predicts $P(y =$ "cat"$) = \hat{p} = 0.6$ for all items (i.e., $\mathcal{U} = \mathcal{X}$). This classifier is perfectly calibrated in terms of aggregate calibration, but it is uninformative about any individual animal. If cats and birds are not separable in the feature space, this may be the best one can do. However, if the items have a feature such as "number of legs", then two subregions—$\mathcal{U}_1$ for animals with two legs and $\mathcal{U}_2$ for animals with four legs—can be defined with true conditional probabilities of 1 (for "cats") and 0 (for "birds"). This heterogeneity is hidden in the classifier's predictions.

This extreme situation (complete HH) could happen for a number of reasons (majority-class classifier, classifier only trained on cats, etc.). More commonly, any classifier may have one or more subregions $\mathcal{U}$ in its predicted probabilities that likewise obscure informative heterogeneity, due to model misspecification, an overly constrained hypothesis space, over-regularization, or data shift. Detecting HH can alert the practitioner to limitations of the classifier. While global methods that map $\hat{p}$ to $\hat{q}$ cannot address HH, local calibration could model the subregions separately and assign $\hat{q}_i$ differently for each $\mathcal{U}_i$.

### 3.2 Detecting Hidden Heterogeneity

Algorithm 1 provides a method to compute the *detectable* hidden heterogeneity for a region $\mathcal{U} \subseteq \mathcal{C}$ given a labeled calibration data set $\mathcal{C}$ sampled from the same distribution as the test set. HH is calculated as the potential improvement (compared to the original classifier) achieved by training a specialized classifier on only the items in $\mathcal{U}$.

In step 1, we define item $x_t$'s probability neighborhood $\mathcal{U}_t$ to contain calibration items that are close to $x_t$ in the probability simplex $\Delta_{K-1}$. More precisely, $\mathcal{U}_t$ contains the items within radius $r$ of item $x_t$ in $\Delta_{K-1}$. There is no *a priori* best choice for $r$, but to obtain reliable HH estimates, one should choose $r$ such that no set $\mathcal{U}_t$ is excessively small. We employ the standard choice of Hellinger distance $D_H$ (Equation 1) to

calculate the distance between probability vectors $\hat{p}_i$ and $\hat{p}_j$. Hellinger distance is the probabilistic equivalent of Euclidean distance, and it is more suitable here than KL divergence, which is not symmetric.

$$D_H(\hat{p}_i, \hat{p}_j) = \frac{1}{\sqrt{2}} \sqrt{\sum_{k=1}^{K} \left( \sqrt{\hat{p}_i[k]} - \sqrt{\hat{p}_j[k]} \right)^2}. \tag{1}$$

Conveniently, the Hellinger distance can be expressed as the Euclidean norm of the difference of the element-wise square root of each probability vector (Krstovski et al., 2013):

$$D_H(\hat{p}_i, \hat{p}_j) = \frac{1}{\sqrt{2}} \left\| \sqrt{\hat{p}_i} - \sqrt{\hat{p}_j} \right\|_2. \tag{2}$$

This in turn allows the use of efficient methods (e.g., k-d tree) for populating neighborhood $\mathcal{U}_t$.

For each test item $x_t$, a new (local) classifier $g_t$ is trained using only the nearby calibration items in $\mathcal{U}_t$ (step 2). This classifier $g_t$ can be any classifier type. We employed an ensemble method that can perform internal generalization estimates without an additional validation set. We trained a bagged ensemble of 50 decision trees with no depth limit and no limit on the number of features searched for each split. We used out-of-bag error to determine how much pruning to employ to achieve good generalization and avoid overfitting to the calibration set. We searched over 7 values of the $\alpha$ pruning complexity parameter, evenly spaced between 0.0 (no pruning) and 0.03, as input to the minimal cost-complexity pruning method (Breiman et al., 1984).

Finally (step 6), we calculate HH for $\mathcal{U}_t$ by comparing the Brier score (Brier, 1950) of the original predictions by model $f$ on $\mathcal{U}_t$ (step 3) with those generated by the local model $g_t$ (step 4) using true labels $Y_{\mathcal{U}_t}$ (step 5):

$$HH_{\mathcal{U}_t} = \mathrm{Brier}(f(\mathcal{U}_t), Y_{\mathcal{U}_t}) - \mathrm{Brier}(g_t(\mathcal{U}_t), Y_{\mathcal{U}_t}), \tag{3}$$

where the Brier score is the mean squared error between predictions $\hat{p}_i[k] \in [0, 1]$ and labels $y_i$, for $N$ items and $K$ possible classes:

$$\mathrm{Brier}(\hat{p}, Y) = \frac{1}{N} \sum_{i=1}^{N} \sum_{k=1}^{K} \left( \hat{p}_i[k] - \mathbb{1}(y_i = k) \right)^2. \tag{4}$$

We enforce the condition that $g_t$ is no worse than $f$ by clipping $HH_{\mathcal{U}_t}$ to 0. Regions with large HH values provide both a warning that global calibration methods may not perform well and an opportunity for local specialization by using item similarity during calibration.

## 4 Similarity-Weighted Calibration

We propose to improve point-wise calibration by leveraging information in feature space as well as the uncalibrated probabilities $\hat{p}_i$. Given test item $x_t$, the goal is to estimate well-calibrated $\hat{q}_t[k] = P(y = k|x_t)$ for each class $k \in \{1 \ldots K\}$. Similarity-Weighted Calibration (SWC) is described in Algorithm 2. Let

$$s(t, i) = \mathrm{sim}([x_t, \hat{p}_t], [x_i, \hat{p}_i]) \in [0, 1]$$

be the similarity between item $x_t$ and item $x_i$ measured in the augmented space $\mathcal{X}^+$, where $[a, b]$ is the concatenation of vectors $a$ and $b$. A similarity of 1 is perfect identity. The investigator chooses how best to measure similarity in this space. Use of $\mathcal{X}^+$ enables calibration to benefit from information encoded by the classifier ($\hat{p}$) as well as item position in feature space ($x$). Further, a supervised similarity measure can learn the relative importance of each component for the problem at hand. We employ such a measure: the random forest *proximity function* (RFprox). RFprox trains a random forest on a labeled data set and defines the similarity between items $x_i$ and $x_j$ as the fraction of times they are assigned to the same leaf in each tree of the ensemble (Breiman, 2001; Cutler et al., 2012). Effectively, the random forest encodes a "kernel" defined by those weights (leaf co-occurrences) (Hastie et al., 2009). We employ the calibration data to learn the relevant RFprox measure using a random forest with 100 trees, no depth limit, and considering a random set of $\sqrt{d}$ features for each split, given $d$ total features. Note that defining sim() using standard

---

**Algorithm 2** Similarity-Weighted Calibration (SWC)

**Input**: Test item $x_t$, calibration data $x_i \in \mathcal{C}$ and labels $y_i$
**Output**: Calibrated probabilities $\hat{q}_t[k], \forall k$

 1: Collect model predictions for item $x_t$: $\hat{p}_t[k]$ for $k \in \{1 \ldots K\}$.
 2: Collect model predictions for the calibration set: $\hat{p}_i[k]$ for $x_i \in \mathcal{C}, k \in \{1, \ldots, K\}$.
 3: Compute pairwise similarity as $s(t,i)$ for $x_i \in \mathcal{C}$.
 4: Compute $\hat{q}_t[k] = \frac{1}{\sum_i s(t,i)} \sum_i s(t,i)\mathbb{1}(y_i = k)$ for $k \in \{1, \ldots, K\}$ (Eqn. 5).

---

kernels, such as the Gaussian kernel, over $\mathcal{X}^+$ would impose a fixed weighting on the $x$ and $\hat{p}$ components. A potential direction for future research would be to apply multiple kernel learning (Gönen & Alpaydın, 2011) to optimally combine separate kernels for $x$ and $\hat{p}$.

SWC computes the similarity of $x_t$ to every item in the calibration set (step 3) and uses this information to replace $\hat{p}$ with a similarity-weighted combination of labels from the calibration set (step 4).

$$\hat{q}_t[k] = \frac{1}{\sum_i s(t,i)} \sum_i s(t,i)\mathbb{1}(y_i = k). \tag{5}$$

The similarity-based approach to calibration enables local specialization within the data set, but it does not directly make use of the calculated HH. We also developed the SWC-HH algorithm, which filters the calibration set to restrict which items are used to generate $\hat{q}$. HH, which is computed separately for each test item $x_t$ (Algorithm 1), is employed as an additional filter for calibration. In step 4, SWC-HH only includes items with a similarity to item $x_t$ of at least $\frac{1}{2}HH_{\mathcal{U}_t}$. Note that the maximum value for HH is 2.0 since it is the difference in Brier scores, clipped to 0.0, and each Brier score ranges between 0.0 and 2.0. Dividing by 2.0 normalizes the threshold to the range 0.0–1.0, making it suitable as a similarity threshold.

## 5 Experimental Results

We conducted experiments with a variety of classifiers and data sets to compare local and global calibration methods and to determine the role that hidden heterogeneity plays. Our hypotheses were that (1) local, similarity-based calibration is more effective at reducing Brier score than global calibration methods, (2) the amount of improvement correlates with the hidden heterogeneity score, and (3) calibration support can serve as an indicator of per-item calibration quality. Our implementations of SWC, SWC-HH, and other calibration methods, along with scripts to replicate the experiments, are available at `https://github.com/wkiri/simcalib`.

### 5.1 Methodology

We assessed calibration methods for six tabular data classifiers as implemented in the scikit-learn Python library (Pedregosa et al., 2011), including a decision tree with `min_samples_leaf = 10` (DT), a random forest with 200 trees (RF), an ensemble of 200 gradient-boosted trees (GBT), a linear support vector machine (SVM), a Gaussian kernel ($\gamma = \frac{1}{d\,var(X)}, C = 1.0$) support vector machine (RBFSVM), and a Naive Bayes classifier (NB). Any parameters not explicitly mentioned were set to their default values. We also conducted experiments with pre-trained deep neural networks for classifying images (details in Section 5.4).

**Data sets.**  We analyzed four tabular and two image data sets:

- moons: a synthetic 2D data set with two partially overlapping classes, chosen to enable visualization of classifier outputs and hidden heterogeneity in feature space. Observations were generated using the scikit-learn `make_moons()` function with `noise` set to 0.3 and `random_state` set to 0.
- letter (letter recognition): a 26-class data set of capital letters from the English alphabet represented by 16 statistical and geometrical features that describe the image of the letter (Frey & Slate, 1991), available at `https://archive.ics.uci.edu/ml/datasets/letter+recognition`.

- mnist: the MNIST handwritten digit data set (LeCun et al., 1998) composed of 28x28 pixel ($d = 784$) images containing a digit from 0 to 9. We used the data set as provided by OpenML; the original source is `http://yann.lecun.com/exdb/mnist/`. In addition to the 10-class data set (mnist10), we created several binary subsets consisting only of two digits, such as "4" and "9" (mnist-4v9). This data set is "tabular" (1-d feature vector); no 2D image structure is preserved.
- fashion-mnist: grayscale images of clothing and accessories (10 classes) that was designed to be more challenging than the MNIST data set yet have the same dimensionality (28x28, $d = 784$) (Xiao et al., 2017), available at `https://github.com/zalandoresearch/fashion-mnist`.
- CIFAR-10: 60,000 images ($64 \times 64$ pixels) labeled into 10 distinct classes; the test set contains 10,000 images (Krizhevsky, 2009)
- CIFAR-100: a disjoint set of 60,000 images labeled into 100 different classes (50k train, 10k test)

For tabular data sets, we randomly sampled 10,000 items and for each trial randomly split them into 500 train, 500 test, and 9000 for a calibration pool. For the "mnist10" and "letter" data sets, we used 1000 items each for training and test, due to their large number of classes (10 and 26, respectively). We created a series of nested calibration sets of size { 50, 100, 200, 500, 1000, 1500, 2000, 2500, 3000 } to assess the data efficiency of each calibration method. For image data sets, in each trial we generated a class-stratified random split of the standard test set into 5000 test items and reserved the remainder as the calibration set. We report average performance across 10 trials along with the standard error for the observed values.

**Calibration methods.** We compared three similarity-based calibration methods (SBA-10, SWC, and SWC-HH) to standard calibration methods including Platt scaling (Platt, 1999), histogram binning (Zadrozny & Elkan, 2001), and isotonic regression (Zadrozny & Elkan, 2002). SBA-10 employed Euclidean distance to identify the nearest neighbors, while SWC and SWC-HH used the RFprox similarity measure. When computing hidden heterogeneity, we used a probability radius of $r = 0.1$ in the probability simplex.

Platt scaling optimizes the negative log-likelihood of predictions against the target probabilities rather than discrete $\{0, 1\}$ labels (Platt, 1999; Niculescu-Mizil & Caruana, 2005). With $n_+$ as the number of calibration items in the positive class and $n_-$ as the number of negative items, the target probabilities are $y_+ = \frac{n_+ + 1}{n_+ + 2}$ and $y_- = \frac{1}{n_- + 2}$ . For multi-class problems, we applied temperature scaling (Guo et al., 2017). For classifiers that output probabilities instead of logits, we first transformed $\hat{p}_i$ into logits as $z_i[k] = \ln \frac{\hat{p}_i[k]}{1 - \hat{p}_i[k]}$. To avoid dividing by zero (or taking its logarithm), we clipped $\hat{p}_i[k]$ to the range $[\epsilon, 1 - \epsilon]$, where $\epsilon = 1 \times 10^{-12}$. We applied the histogram binning method implemented by Kumar et al. (2019)[1] and followed their recommendation to use equal-mass bins (100 bins).

**Metrics.** We employ Brier score (Equation 4) to measure point-wise prediction quality, following earlier researchers (e.g., Zadrozny & Elkan, 2001; 2002; Niculescu-Mizil & Caruana, 2005). It has several advantages over a commonly used calibration metric called the Expected Calibration Error (ECE) (Naeini et al., 2015), which assigns predictions to bins to assess aggregate calibration. ECE can be trivially minimized to 0 by always predicting the empirical average probability of a given class, yielding perfectly calibrated but uninformative predictions (Kull et al., 2017; Widmann et al., 2019; Ovadia et al., 2019). The Brier score incorporates not just calibration error (or "reliability") but also "resolution" or sharpness, which rewards predictions that do more than predict the average probability (Ferro & Fricker, 2012). In addition, ECE is sensitive to the number and choice of bins (Vaicenavicius et al., 2019; Kumar et al., 2019; Patel et al., 2021), it exhibits undesirable edge effects (discontinuities at bin boundaries), and it only assesses calibration of the predicted class. Brier score characterizes prediction quality across all classes, and it avoids artificial discretization and edge effects, since it does not employ binning.

## 5.2 Similarity-based calibration for tabular data sets

To test our first hypothesis, we compared similarity-based calibration to global methods. Figure 1 shows Brier score as a function of available calibration data for the binary classification task of distinguishing two

---

[1]Available at `https://github.com/p-lambda/verified_calibration`.

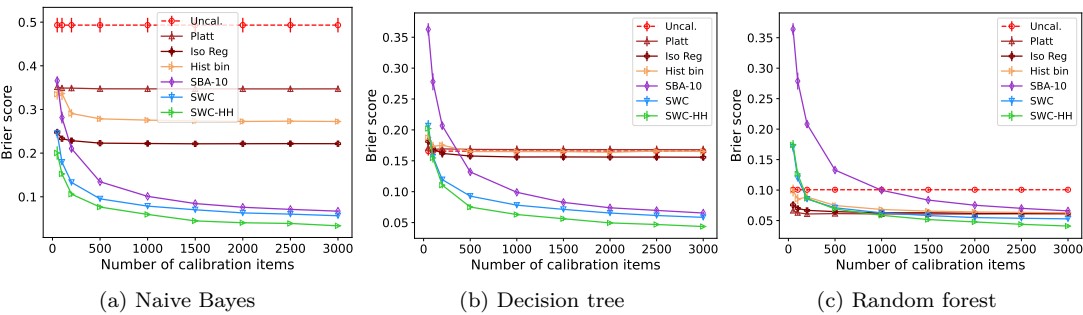

(a) Naive Bayes                (b) Decision tree                (c) Random forest

Figure 1: Calibration performance for three classifiers on the MNIST 4-vs-9 data set. Each plot shows Brier score (lower is better) with increasing calibration set size. Error bars show one standard error over 10 trials.

handwritten MNIST digits ("4" vs. "9"). The uncalibrated predictions yielded different starting Brier scores for each classifier (red dashed lines). Platt scaling and isotonic regression improved the Brier score for the Naive Bayes (NB) and random forest (RF) classifiers but only marginally for the decision tree (DT). No further improvements were achieved beyond 500 calibration items. Similarity-based calibration (SBA-10, SWC, and SWC-HH) generally did not perform well with small calibration sets but achieved much larger benefits for NB and DT when at least 500 items were used for calibration, and Brier score continued to improve as more calibration data was provided. The random forest, which had the best initial Brier score and therefore less room for improvement, showed an advantage for similarity-based calibration after at least 1500 items were used. SWC-HH yielded a clear advantage at all calibration set sizes for NB and DT, and it also provided a small advantage over SWC for RF with calibration set sizes of at least 1500 items.

SWC and SWC-HH consistently out-performed SBA-10. Since RFprox uses labels to learn the similarity measure, it can elevate the importance of individual features in $\mathcal{X}^+$ (like $\hat{p}[k]$) by placing them higher in individual decision trees within its ensemble. SWC and SWC-HH also include evidence from the entire calibration set rather than only the nearest neighbors. Importantly, SBA-10 showed almost no difference in Brier score for different classifiers, effectively ignoring their individual differences in $\hat{p}$. MNIST is represented by 784 features, so the addition of two dimensions in $\mathcal{X}^+$ has little effect. In contrast, the fact that SWC obtained different absolute results for each classifier indicates that RFprox effectively leveraged the $\hat{p}$ features. Experimental results for all classifiers and all tabular data sets, reporting Brier score and accuracy results, are provided in Appendix A (Figures 7 and 8).

### 5.3 Similarity-based calibration exploits hidden heterogeneity

Our second hypothesis was that HH helps explain why and when similarity-based calibration is effective. We examined Brier score improvement across a large combination of data sets, classifiers, and random trials. We found that large HH can be present even in the absence of domain shift, which creates a large opportunity for local calibration. Figure 2 shows the improvement (reduction) in Brier score after calibration as a function of the average HH across the test set. The four binary data sets were MNIST "1" vs. "7" (relatively easy), "4" vs. "9" and "3" vs. "8" (intermediate), and "3" vs. "5" (difficult). The three multi-class data sets were "mnist10", "fashion-mnist", and "letter". We compared Platt (for binary) or temperature scaling (multi-class) calibration to SWC and SWC-HH for six classifiers, using 10 trials per combination of data set and classifier. SWC and SWC-HH achieved Brier score improvements that correlate strongly with the amount of measured HH. The relationship was far weaker for the global Platt/temperature scaling methods, which cannot detect or exploit HH. These results show that HH, which can be computed prior to calibration, is a useful diagnostic indicator that can guide the choice of calibration strategy. When HH is high, it is advisable to employ a similarity-based calibration method like SWC. When HH is low or there is not much calibration data, global methods such as Platt scaling are recommended.

To better understand how SWC and SWC-HH improve Brier score, we visualize the calibration process in Figure 3 for the two-dimensional "moons" data set. Each row corresponds to results for a different classifier

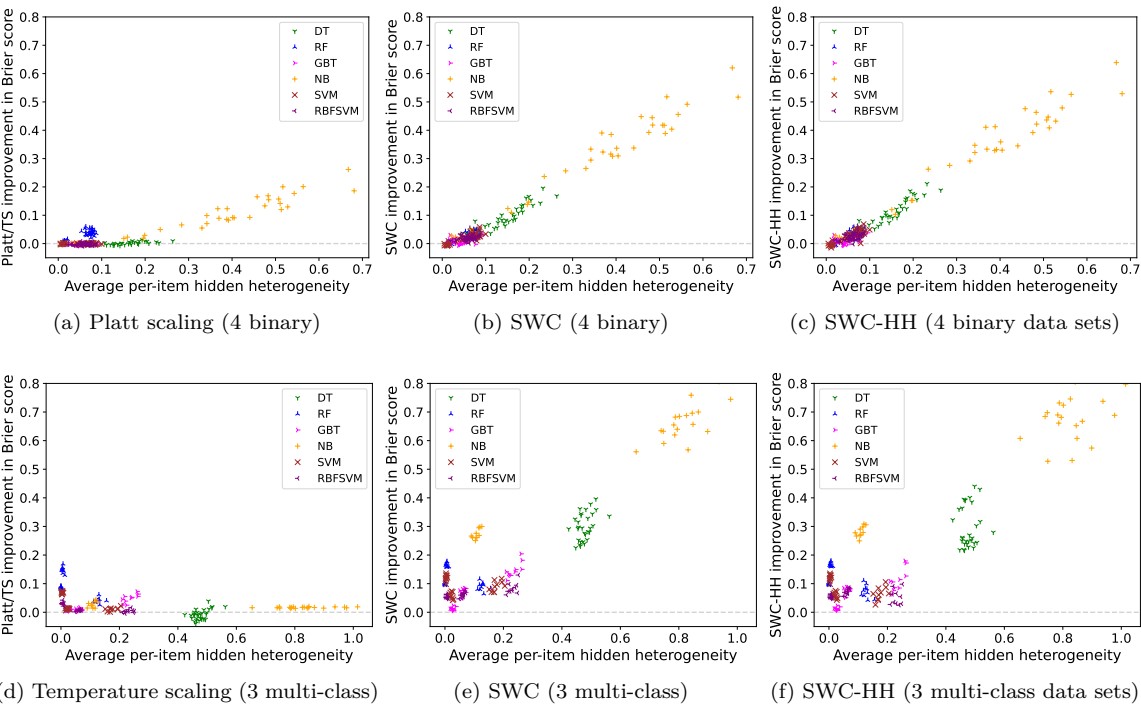

Figure 2: Brier score improvement versus average hidden heterogeneity for four binary (top row) and three multi-class (bottom row) tabular data sets, for six classifier types and 10 random trials.

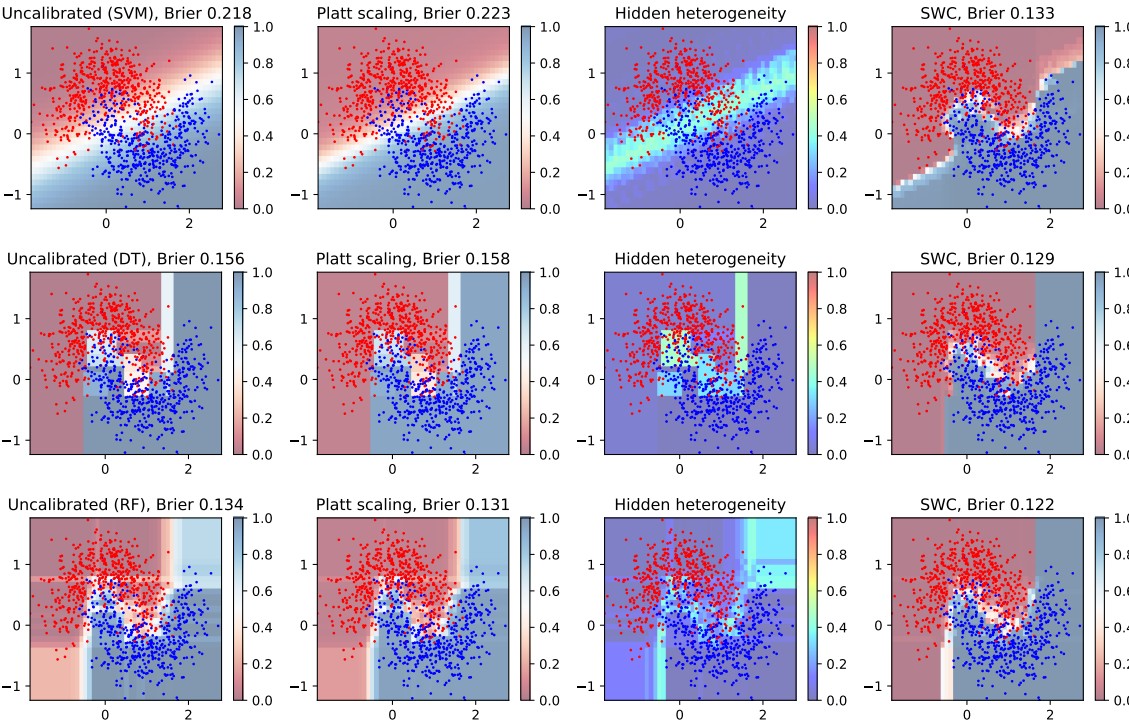

Figure 3: Visualization of uncalibrated predictions (first column) for the "moons" data set, Platt scaling (second column), and SWC (fourth column). The third column shows hidden heterogeneity values that highlight areas of potential improvement, which align with SWC improvements.

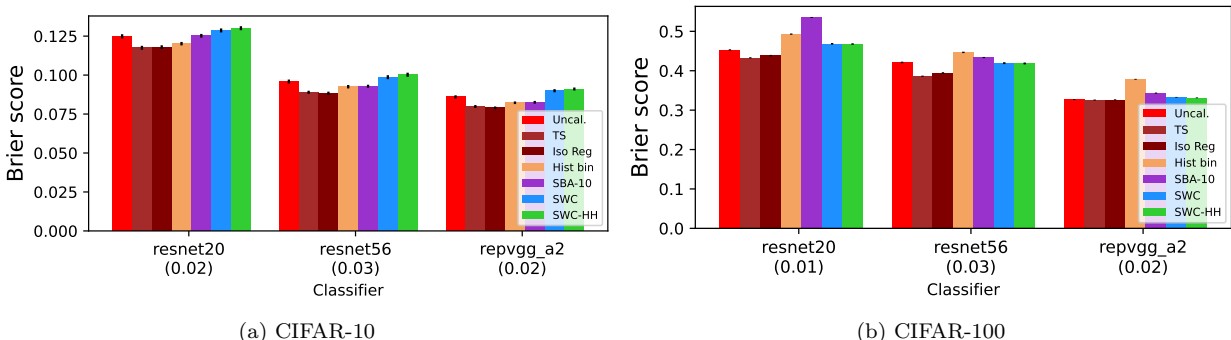

Figure 4: Calibration performance on CIFAR-10 and CIFAR-100 using three pre-trained neural networks over 10 trials (error bars show standard error). HH is shown in parentheses.

(linear SVM, decision tree, and random forest). The third column shows the computed HH values. The classifiers were trained on 500 points, calibrated using 1000 points, and Brier score evaluated on 500 points. The linear SVM cannot model the nonlinear decision boundary very well. The diagonal region where the classes are mixed yet separable in the feature space has a high value for HH. When SWC is applied, it is the $\hat{p}$ values in this band that receive the biggest modifications. These changes reduce (improve) the Brier score from 0.218 to 0.133. Platt scaling increases (worsens) the Brier score slightly. The decision tree (second row of Figure 3) exhibits less hidden heterogeneity on the same data set, because it is able to model the nonlinear decision boundary more effectively. SWC improves the Brier score from 0.156 to 0.129. Finally, the random forest (bottom row of Figure 3) exhibits more areas with hidden heterogeneity, due to overly conservative predictions in the upper right and lower left areas. SWC creates smoother regions as the calibration data informs updates to the posterior probabilities and improves the Brier score from 0.134 to 0.122.

Importantly, the difference in results for the rightmost column in Figure 3 demonstrates that SWC adapts (calibrates) most where the classifier exhibits hidden heterogeneity, yielding a result that is customized to the original classifier and more flexible than global calibration.

## 5.4 Similarity-based calibration for image classifiers

We also conducted calibration experiments with the CIFAR-10 and CIFAR-100 data sets (Krizhevsky, 2009) using three pre-trained neural networks of increasing complexity[2]. ResNet20 (He et al., 2016) has 20 layers and 0.27M parameters, ResNet56 (He et al., 2016) has 56 layers and 0.85M parameters, and RepVGG_A2 (Ding et al., 2021) has 22 layers and 25.49M parameters. For these data sets, the similarity measure *sim* used by SWC and SWC-HH operates in the latent space learned by each network. Specifically, we used the output activations of the `avgpool` (for ResNet models) and `gap` (for RepVGG_A2) layers as a feature vector (dimensionality 64, 64, and 1408 respectively). We again used a learned RF proximity function to compute similarity in this space. We found that a probability radius of 0.05 yielded reasonably sized neighborhoods for computing HH.

Improvements for CIFAR-10 and CIFAR-100 are evident as more complex neural networks are trained; the Brier score consistently decreases from ResNet20 to ResNet56 to RepVGG_A2 (see Figure 4). However, HH values were very low for all three networks $(0.01-0.03)$, leaving little room for improvement with local calibration. Indeed, we found that global calibration (temperature scaling or isotonic regression) yielded the best results for these data sets. Consistent with the results on tabular data shown in section 5.2, calculating HH in advance provides guidance as to which method will be most useful.

There is room for additional improvement. Recent studies of deep network latent spaces suggest that distances computed in neural network latent spaces often do not work well. For example, nearest neighbor classifiers using latent space distances perform substantially worse than the standard multinomial logistic

---

[2]Pre-trained models were obtained from `https://github.com/chenyaofo/pytorch-cifar-models` .

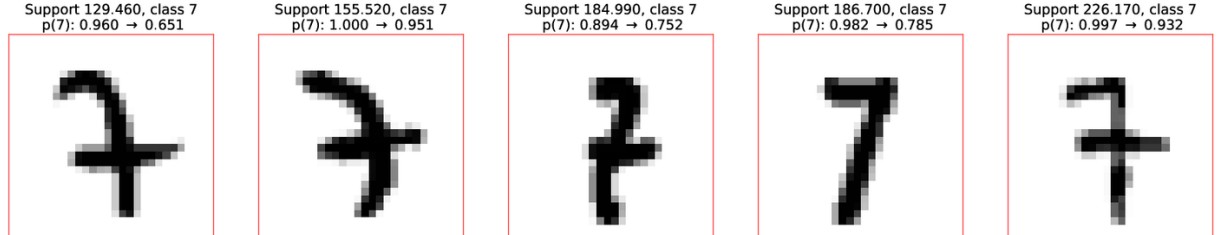

Figure 5: Test items from "mnist-1v7" with the lowest calibration support for a linear SVM classifier. For all five items, SWC *reduced* the confidence of the correct class, instead of increasing it.

regression (softmax) classifiers (Garrepalli, 2022). These latent spaces are also not able to represent dimensions of variation that were poorly sampled in the training data (Dietterich & Guyer, 2022). Likewise, decision tree classifiers do not perform well on these learned representations (Garrepalli, 2022). Since HH is influenced by the data representation, hidden heterogeneity could be more detectable for these data sets using a different representation. Likewise, even if HH is large, similarity-based local calibration may not always be able to improve the Brier score due to limitations of the representation. Employing t-SNE or PCA to reduce dimensionality for similarity calculations (as done by Luo et al. (2022)) could also be beneficial. Exploring the connection between choice of representation and calibration efficacy is an important future direction.

### 5.5 Calibration support highlights calibration data gaps and domain shift

Our third hypothesis is that measuring calibration support, which is a unique capability of similarity-based calibration methods, can provide useful information about the relevance of the calibration set to each item being calibrated. We define the *calibration support $S$* for item $t$ that informs $\Phi(\hat{p}|t)$ as the sum of similarity weights for items drawn from calibration set $\mathcal{C}$:

$$S_{t,\mathcal{C}} = \sum_i s(t, i), x_i \in \mathcal{C}. \tag{6}$$

Identifying items with low values for $S_{t,\mathcal{C}}$ can draw attention to observations that are not well represented by the calibration set. These could be individual outliers or, if there is a large number of such items, they could indicate distribution shift between the calibration and test sets. Low $S_{t,\mathcal{C}}$ values signal the need for more data (or more representative data) to be added to $\mathcal{C}$. While previous studies focus solely on calibration performance as a function of the total calibration set *size*, similarity-based calibration can characterize the *relevance* of the calibration set to individual test items.

Consider a linear SVM trained on 500 MNIST "1" and "7" digits and calibrated using SWC with 3000 digits. For most items in the test set, calibration improves. However, examining items with low calibration support helps us understand failures. Figure 5 shows the five test items (out of 500 total) with the lowest calibration support. Calibration with SWC was detrimental ($\hat{q}[y]$ decreased) for all five. These items are not necessarily ambiguous in an objective sense, but the fact that they have low representation in the calibration set signals (correctly) that the calibrated output $\hat{q}$ may be less reliable.

Likewise, low calibration support can provide a warning when domain shift is present between the calibration and test sets (or any new prediction). We simulated domain shift (covariate shift) by rotating a subset of the test items 90 degrees counter-clockwise. Figure 6(a) shows the distribution of calibration support values without any rotation; the peak value is around 900, and items with low support are rare. With 10% of the test set rotated (Figure 6(b)), the overall histogram changes a little, and the rotated items (orange bars) tend to have low calibration support, signaling a need for inspection. With 50% (partial domain shift) rotated (Figure 6(c)), distinct populations for the rotated and unrotated items are clear, and with 100% rotated (complete domain shift), the whole histogram shifts to lower values (peak around 300).

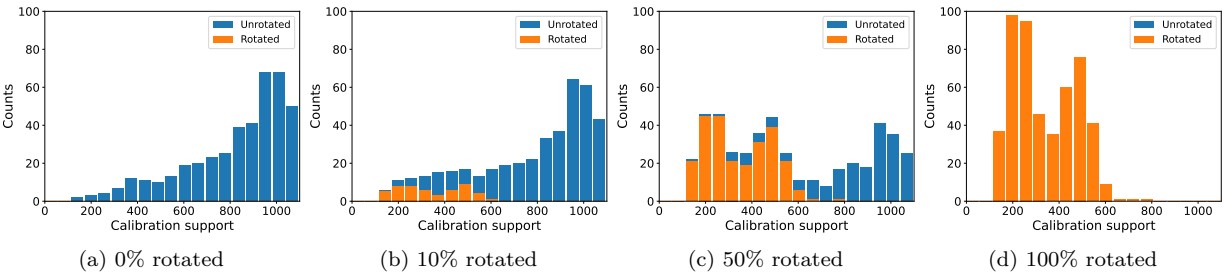

Figure 6: Distribution of calibration support values for 500 test items ("mnist-1v7") classified by a linear SVM with progressively more of the test set items rotated. Bar plots are stacked.

This result suggests that in a deployment setting, it is useful to monitor the calibration support values that are reported by SWC. Knowledge about typical support values (or better, their distribution) could enable early detection of domain shift when new items originate from a changed distribution. That signal indicates that the calibration set, and likely the trained model as well, require revision. Platt, temperature scaling, histogram binning, and other methods provide a fixed mapping $\Phi(\hat{p})$ without regard to the item being calibrated; there is no signal to indicate whether $\Phi$ is still relevant. SWC provides an intrinsic measure of calibration set relevance through the calibration support values obtained by each new item.

## 6 Conclusions, Limitations, and Future work

In this work, we explored the benefits of local, individual calibration for each test item, in contrast to widely used global classifier calibration methods. We identified hidden heterogeneity (HH) as a strong indicator of the need for local calibration, when there are subpopulations within a data set that have the same uncalibrated predicted probability $\hat{p}$ yet require different corrections to achieve a well-calibrated probability $\hat{q}$. We provided a method for calculating HH before calibration to inform selection of the calibration method. Experiments with tabular data sets and diverse machine learning classifiers indicate that local calibration improves Brier score in proportion to the average hidden heterogeneity (HH) value in the data set. We highlight this finding as an important step towards not only correcting miscalibration but also explaining and understanding it. When HH is very low (as we found with several deep neural networks), or little calibration data is available, global methods such as temperature scaling are sufficient, but otherwise, local calibration is preferred. On the other hand, because local calibration has far more degrees of freedom than parametric, global methods, it tends to require more calibration data. If calibration data is scarce, global methods may be preferred.

We proposed a similarity-based approach to local calibration (SWC) that weights evidence from the calibration set according to its similarity to the test item in an "augmented" feature space that includes both the input features and the predicted class probabilities. This concept goes beyond prior work such as Similarity-Binning Averaging (SBA-10), which calibrates (without weighting) using the 10 nearest neighbors based on Euclidean distance in the augmented feature space (Bella et al., 2009). In most cases, we found that SWC out-performs SBA-10. Additional benefits can be obtained by incorporating HH directly into the SWC algorithm (SWC-HH). A final and unique benefit of similarity-based calibration is that the explicit measurement of calibration support can serve to warn when a given test item lacks good representation in the calibration set. This can also be an indicator when distribution shift or domain shift is present.

**Limitations: Runtime.** The computational cost of local calibration methods tends to be higher than that of global methods, since each item is independently modeled rather than constructing a single model to apply to all items. However, this also means that calibration can be conducted lazily, as needed, given a similarity measure. The computation of hidden heterogeneity requires (1) the identification of an item's nearest neighbors in the probability simplex $\Delta_{K-1}$, which can be costly with a large number of classes, and (2) training a specialized classifier to estimate the potential Brier score improvement (Equation 3).

**Limitations: Preservation of accuracy.**   SWC and SWC-HH are not rank-preserving calibration methods. This means that in addition to modifying the calibration properties of the predictions, they can also change the predicted class and therefore the accuracy of predictions. Improvements in accuracy are reflected in improved Brier scores. Temperature scaling, in contrast, does preserve rank and accuracy because, without a bias term, it cannot move items to the other side of the decision threshold. Some researchers favor rank-preserving methods (Zhang et al., 2020; Patel et al., 2021), since they seek to improve calibration without sacrificing accuracy. However, this constraint also prevents them from *increasing* accuracy, which is an outcome available to Platt scaling (via its bias term), histogram binning, SWC, etc. On the whole, we agree with Bella et al. (2013) that there is no need to preserve item rankings given the opportunity to improve both accuracy and calibration. However, we acknowledge that in some applications, there could be a need to choose a rank-preserving calibration method to ensure accuracy is unchanged (up or down) for user acceptance (Srivastava et al., 2020).

**Future work.**   There are several important directions for future work. It is possible that within the same data set, some items are best calibrated with global methods while others (where HH is present) benefit from local calibration. A hybrid method that selectively applies global/local calibration, or some combination of the two, for each test item could potentially out-perform either one. For a given problem, alternative choices for data representation and similarity measures could yield additional improvements. In addition, SWC is well designed to naturally accommodate domain shift, if the calibration data set is drawn from the shifted distribution. Sampling bias in the training set, whether intentional or not, induces a particular kind of domain shift that is especially important to address to meet fairness goals when predictions are made in a deployment setting.

### Author Contributions

KW: Problem refinement, algorithm development, implementation, experimentation, data analysis, and writing.
TD: Initial problem formulation, critical feedback, assisting with writing.

### Acknowledgments

This material is based upon work supported by the Defense Advanced Research Projects Agency (DARPA) under Contract No. HR001119C0112. Any opinions, findings and conclusions or recommendations expressed in this material are those of the authors and do not necessarily reflect the views of the DARPA. The authors thank the reviewers and the editor for valuable discussions that significantly improved the paper.

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

# A    Appendix

This appendix provides experimental results for seven data sets and six classifiers, comparing similarity-based calibration to other methods. In these experiments, we randomly sampled 10,000 items from each data set and randomly split them into 500 train (for binary problems) or 1000 train (for multi-class problems), 500 test, and 9000 for a calibration pool.

## A.1    Binary tabular data

Figure 7 presents results across all classifiers and calibration methods for the four binary MNIST data sets after 3000 calibration items were employed. Complete numeric results are shown in Tables 1 to 4. Classifiers appear in order of improving (decreasing) Brier score for the uncalibrated predictions.

Platt scaling improved performance for the Naive Bayes and random forest classifiers, but it yielded little benefit for the others. Isotonic regression sometimes provided additional improvements, primarily for Naive Bayes, and usually out-performed histogram binning. In contrast, similarity-based methods were highly effective for all classifiers in reducing Brier score.

SWC-HH consistently provided the best results. The SBA-10 approach described by Bella et al. (2009) weights all ten neighbors equally. We learned (Ferri, personal communication, Oct. 29, 2022) that the SBA authors have subsequently employed weighting in the averaging process so that each neighbor contributes inversely to its distance from the item to be calibrated. We included the weighted variation in our experiments, denoted as "SBAW-10". Weighting provides slight advantages over SBA-10 in some cases, but SWC/SWC-HH yielded the best results. We can view SBAW as an intermediate choice between SBA and SWC, as it adopts the distance weighting of SWC but not the other aspects (supervised distance metric, HH filtering) that make SWC-HH the strongest method overall. As noted earlier, and most evident in Tables 1 to 4, both SBA and SBAW generate nearly identical results for all classifiers, because they are dominated by feature space distance, and the classifier's initial predictions have little influence. In contrast, SWC and SWC-HH adapt to each classifier's individual limitations (hidden heterogeneity). SWC/SWC-HH improvements correlate with the average HH value, shown in parenthesis under the x axis labels. In addition, similarity-based calibration also increased test accuracy (see right column of Figure 7 and subtables in Tables 1 to 4). SWC-HH consistently achieved the highest accuracy.

Table 1: Results for mnist-1v7 ($n_{cal} = 3000$, 10 trials). The best result(s) for each model (within 1 standard error, shown as a subscript) are in bold.

| | | | | Brier score | | | | |
|---|---|---|---|---|---|---|---|---|
| Model | Uncal. | Platt | Iso Reg | Hist bin | SBA-10 | SBAW-10 | SWC | SWC-HH |
| DT | $0.0678_{0.006}$ | $0.0671_{0.005}$ | $0.0650_{0.006}$ | $0.0913_{0.009}$ | $\mathbf{0.0203}_{0.003}$ | $\mathbf{0.0198}_{0.003}$ | $0.0226_{0.002}$ | $\mathbf{0.0207}_{0.002}$ |
| NB | $0.0452_{0.004}$ | $0.0442_{0.004}$ | $0.0423_{0.003}$ | $0.0674_{0.022}$ | $0.0200_{0.003}$ | $0.0195_{0.003}$ | $\mathbf{0.0165}_{0.002}$ | $\mathbf{0.0168}_{0.002}$ |
| RBFSVM | $0.0411_{0.003}$ | $0.0455_{0.004}$ | $0.0365_{0.002}$ | $0.0358_{0.002}$ | $0.0201_{0.003}$ | $\mathbf{0.0197}_{0.003}$ | $\mathbf{0.0189}_{0.002}$ | $\mathbf{0.0172}_{0.003}$ |
| GBT | $0.0380_{0.004}$ | $0.0390_{0.004}$ | $0.0282_{0.003}$ | $0.0284_{0.002}$ | $\mathbf{0.0203}_{0.003}$ | $\mathbf{0.0198}_{0.003}$ | $0.0205_{0.002}$ | $\mathbf{0.0191}_{0.002}$ |
| RF | $0.0246_{0.002}$ | $\mathbf{0.0170}_{0.002}$ | $\mathbf{0.0173}_{0.002}$ | $\mathbf{0.0182}_{0.002}$ | $0.0202_{0.003}$ | $0.0197_{0.003}$ | $\mathbf{0.0182}_{0.002}$ | $\mathbf{0.0171}_{0.002}$ |
| SVM | $\mathbf{0.0155}_{0.002}$ | $\mathbf{0.0157}_{0.002}$ | $\mathbf{0.0148}_{0.001}$ | $\mathbf{0.0155}_{0.001}$ | $0.0200_{0.003}$ | $0.0196_{0.003}$ | $\mathbf{0.0149}_{0.002}$ | $\mathbf{0.0154}_{0.002}$ |
| | | | | Accuracy | | | | |
| Model | Uncal. | Platt | Iso Reg | Hist bin | SBA-10 | SBAW-10 | SWC | SWC-HH |
| DT | $0.9586_{0.004}$ | $0.9592_{0.004}$ | $0.9614_{0.004}$ | $0.9484_{0.006}$ | $\mathbf{0.9866}_{0.002}$ | $\mathbf{0.9880}_{0.001}$ | $0.9852_{0.001}$ | $\mathbf{0.9874}_{0.001}$ |
| RBFSVM | $0.9740_{0.003}$ | $0.9738_{0.002}$ | $0.9718_{0.002}$ | $0.9750_{0.002}$ | $0.9868_{0.001}$ | $0.9880_{0.001}$ | $0.9876_{0.002}$ | $\mathbf{0.9906}_{0.001}$ |
| NB | $0.9774_{0.002}$ | $0.9774_{0.002}$ | $0.9774_{0.002}$ | $0.9766_{0.002}$ | $0.9868_{0.001}$ | $0.9880_{0.001}$ | $\mathbf{0.9894}_{0.001}$ | $\mathbf{0.9894}_{0.001}$ |
| GBT | $0.9788_{0.002}$ | $0.9790_{0.002}$ | $0.9808_{0.002}$ | $0.9808_{0.002}$ | $\mathbf{0.9866}_{0.002}$ | $\mathbf{0.9880}_{0.001}$ | $0.9864_{0.001}$ | $\mathbf{0.9878}_{0.001}$ |
| RF | $0.9890_{0.001}$ | $\mathbf{0.9898}_{0.001}$ | $0.9888_{0.001}$ | $0.9892_{0.001}$ | $0.9866_{0.001}$ | $0.9878_{0.001}$ | $0.9894_{0.001}$ | $\mathbf{0.9912}_{0.001}$ |
| SVM | $0.9898_{0.001}$ | $\mathbf{0.9904}_{0.001}$ | $\mathbf{0.9910}_{0.001}$ | $0.9900_{0.001}$ | $0.9868_{0.001}$ | $0.9880_{0.001}$ | $\mathbf{0.9902}_{0.001}$ | $\mathbf{0.9914}_{0.001}$ |

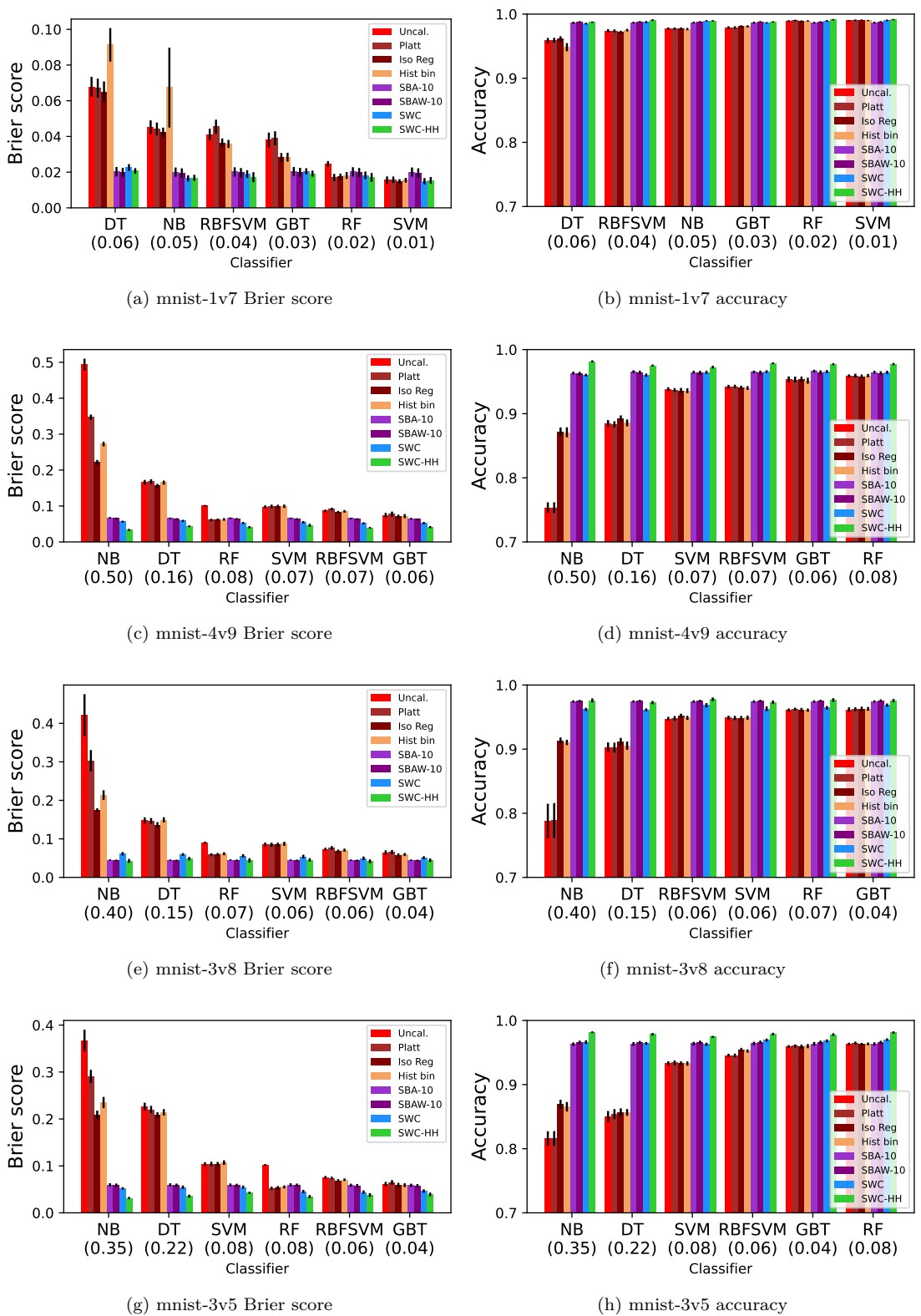

Figure 7: Calibration performance (left) and accuracy (right) for the binary MNIST data sets (10 trials; error bars indicate one standard error). Classifiers are sorted in order of improvement based on the uncalibrated classifier's score, and average HH values are below each classifier.

Table 2: Results for mnist-4v9 ($n_{cal} = 3000$, 10 trials). The best result(s) for each model (within 1 standard error, shown as a subscript) are in bold.

| | Brier score | | | | | | | |
|---|---|---|---|---|---|---|---|---|
| Model | Uncal. | Platt | Iso Reg | Hist bin | SBA-10 | SBAW-10 | SWC | SWC-HH |
| NB | $0.4932_{0.017}$ | $0.3472_{0.007}$ | $0.2216_{0.007}$ | $0.2724_{0.006}$ | $0.0671_{0.002}$ | $0.0653_{0.002}$ | $0.0565_{0.003}$ | $\mathbf{0.0335}_{0.003}$ |
| DT | $0.1657_{0.007}$ | $0.1679_{0.007}$ | $0.1558_{0.006}$ | $0.1651_{0.006}$ | $0.0655_{0.002}$ | $0.0638_{0.002}$ | $0.0585_{0.003}$ | $\mathbf{0.0436}_{0.003}$ |
| RF | $0.1005_{0.002}$ | $0.0609_{0.003}$ | $0.0619_{0.003}$ | $0.0627_{0.004}$ | $0.0659_{0.002}$ | $0.0642_{0.002}$ | $0.0528_{0.003}$ | $\mathbf{0.0410}_{0.003}$ |
| SVM | $0.0975_{0.004}$ | $0.0988_{0.005}$ | $0.0984_{0.005}$ | $0.0993_{0.005}$ | $0.0655_{0.002}$ | $0.0638_{0.002}$ | $0.0547_{0.004}$ | $\mathbf{0.0462}_{0.004}$ |
| RBFSVM | $0.0863_{0.004}$ | $0.0912_{0.004}$ | $0.0827_{0.004}$ | $0.0851_{0.004}$ | $0.0652_{0.002}$ | $0.0636_{0.002}$ | $0.0518_{0.003}$ | $\mathbf{0.0393}_{0.002}$ |
| GBT | $0.0741_{0.007}$ | $0.0773_{0.007}$ | $0.0702_{0.006}$ | $0.0717_{0.006}$ | $0.0647_{0.002}$ | $0.0630_{0.002}$ | $0.0522_{0.003}$ | $\mathbf{0.0413}_{0.003}$ |
| | Accuracy | | | | | | | |
| Model | Uncal. | Platt | Iso Reg | Hist bin | SBA-10 | SBAW-10 | SWC | SWC-HH |
| NB | $0.7534_{0.008}$ | $0.7534_{0.008}$ | $0.8720_{0.007}$ | $0.8710_{0.008}$ | $0.9634_{0.002}$ | $0.9624_{0.003}$ | $0.9602_{0.002}$ | $\mathbf{0.9816}_{0.002}$ |
| DT | $0.8846_{0.005}$ | $0.8834_{0.005}$ | $0.8928_{0.005}$ | $0.8856_{0.005}$ | $0.9656_{0.002}$ | $0.9642_{0.003}$ | $0.9600_{0.003}$ | $\mathbf{0.9752}_{0.001}$ |
| SVM | $0.9382_{0.003}$ | $0.9368_{0.003}$ | $0.9360_{0.004}$ | $0.9358_{0.004}$ | $0.9648_{0.002}$ | $0.9638_{0.003}$ | $0.9646_{0.002}$ | $\mathbf{0.9726}_{0.002}$ |
| RBFSVM | $0.9420_{0.003}$ | $0.9424_{0.003}$ | $0.9406_{0.003}$ | $0.9400_{0.003}$ | $0.9652_{0.002}$ | $0.9642_{0.003}$ | $0.9656_{0.002}$ | $\mathbf{0.9786}_{0.001}$ |
| GBT | $0.9534_{0.005}$ | $0.9532_{0.005}$ | $0.9538_{0.004}$ | $0.9514_{0.005}$ | $0.9664_{0.002}$ | $0.9646_{0.003}$ | $0.9656_{0.002}$ | $\mathbf{0.9772}_{0.002}$ |
| RF | $0.9586_{0.003}$ | $0.9594_{0.003}$ | $0.9584_{0.003}$ | $0.9594_{0.003}$ | $0.9648_{0.002}$ | $0.9632_{0.003}$ | $0.9644_{0.003}$ | $\mathbf{0.9774}_{0.002}$ |

Table 3: Results for mnist-3v8 ($n_{cal} = 3000$, 10 trials). The best result(s) for each model (within 1 standard error, shown as a subscript) are in bold.

| | Brier score | | | | | | | |
|---|---|---|---|---|---|---|---|---|
| Model | Uncal. | Platt | Iso Reg | Hist bin | SBA-10 | SBAW-10 | SWC | SWC-HH |
| NB | $0.4212_{0.054}$ | $0.3029_{0.028}$ | $0.1749_{0.004}$ | $0.2136_{0.013}$ | $0.0451_{0.002}$ | $\mathbf{0.0446}_{0.002}$ | $0.0613_{0.004}$ | $\mathbf{0.0429}_{0.005}$ |
| DT | $0.1477_{0.008}$ | $0.1465_{0.008}$ | $0.1359_{0.008}$ | $0.1494_{0.007}$ | $\mathbf{0.0448}_{0.002}$ | $\mathbf{0.0443}_{0.002}$ | $0.0594_{0.004}$ | $\mathbf{0.0486}_{0.004}$ |
| RF | $0.0902_{0.003}$ | $0.0590_{0.004}$ | $0.0595_{0.004}$ | $0.0614_{0.004}$ | $\mathbf{0.0452}_{0.002}$ | $\mathbf{0.0446}_{0.002}$ | $0.0559_{0.004}$ | $\mathbf{0.0441}_{0.006}$ |
| SVM | $0.0853_{0.005}$ | $0.0848_{0.005}$ | $0.0847_{0.005}$ | $0.0871_{0.005}$ | $\mathbf{0.0450}_{0.002}$ | $\mathbf{0.0444}_{0.002}$ | $0.0539_{0.005}$ | $\mathbf{0.0454}_{0.004}$ |
| RBFSVM | $0.0733_{0.004}$ | $0.0764_{0.005}$ | $0.0682_{0.004}$ | $0.0707_{0.004}$ | $0.0448_{0.002}$ | $0.0443_{0.002}$ | $0.0494_{0.005}$ | $\mathbf{0.0418}_{0.005}$ |
| GBT | $0.0646_{0.006}$ | $0.0657_{0.005}$ | $0.0581_{0.004}$ | $0.0593_{0.004}$ | $\mathbf{0.0446}_{0.002}$ | $\mathbf{0.0441}_{0.002}$ | $0.0508_{0.004}$ | $\mathbf{0.0445}_{0.005}$ |
| | Accuracy | | | | | | | |
| Model | Uncal. | Platt | Iso Reg | Hist bin | SBA-10 | SBAW-10 | SWC | SWC-HH |
| NB | $0.7878_{0.027}$ | $0.7888_{0.027}$ | $0.9132_{0.005}$ | $0.9104_{0.004}$ | $\mathbf{0.9744}_{0.002}$ | $\mathbf{0.9750}_{0.002}$ | $0.9620_{0.003}$ | $\mathbf{0.9760}_{0.003}$ |
| DT | $0.9030_{0.008}$ | $0.9024_{0.008}$ | $0.9116_{0.006}$ | $0.9054_{0.006}$ | $\mathbf{0.9744}_{0.002}$ | $\mathbf{0.9754}_{0.001}$ | $0.9612_{0.002}$ | $0.9726_{0.002}$ |
| RBFSVM | $0.9472_{0.003}$ | $0.9480_{0.004}$ | $0.9522_{0.003}$ | $0.9488_{0.003}$ | $0.9744_{0.002}$ | $0.9754_{0.002}$ | $0.9684_{0.003}$ | $\mathbf{0.9778}_{0.003}$ |
| SVM | $0.9488_{0.003}$ | $0.9486_{0.003}$ | $0.9480_{0.004}$ | $0.9490_{0.003}$ | $\mathbf{0.9744}_{0.002}$ | $\mathbf{0.9754}_{0.002}$ | $0.9630_{0.004}$ | $\mathbf{0.9732}_{0.003}$ |
| RF | $0.9608_{0.003}$ | $0.9624_{0.002}$ | $0.9616_{0.003}$ | $0.9608_{0.003}$ | $0.9744_{0.002}$ | $\mathbf{0.9756}_{0.001}$ | $0.9644_{0.003}$ | $\mathbf{0.9766}_{0.003}$ |
| GBT | $0.9616_{0.004}$ | $0.9620_{0.003}$ | $0.9630_{0.003}$ | $0.9628_{0.003}$ | $\mathbf{0.9744}_{0.002}$ | $\mathbf{0.9758}_{0.002}$ | $0.9686_{0.003}$ | $\mathbf{0.9758}_{0.002}$ |

## A.2 Multi-class tabular data

Figure 8 and Tables 5 to 7 present results for the multi-class data sets "mnist10", "fashion-mnist", and "letter", after 5000 calibration items were employed. Temperature scaling in general only improved Brier score for the tree-based methods (RF, GBT), and not consistently. Histogram binning was again beneficial for the Naive Bayes models, but it often made calibration worse for decision trees. Isotonic regression yielded small additional improvements.

For "mnist10" and "fashion-mnist", similarity-based calibration provided the best results. SWC and SWC-HH out-performed SBA-10. SWC-HH usually improved over SWC, except for the more challenging "fashion-mnist" data set. In this data set, the filtering employed by SWC-HH (to ignore calibration items with insufficient similarity) often resulted in no calibration items remaining. We handle this case by using the single nearest neighbor, even if its similarity is below the threshold. This leads to values for $\hat{q}$ that are based only on one calibration item. In many cases, the single nearest neighbor belongs to the correct class, yielding good accuracy, but when it is from an incorrect class, the Brier score penalty is large. However, the

Table 4: Results for mnist-3v5 ($n_{cal} = 3000$, 10 trials). The best result(s) for each model (within 1 standard error, shown as a subscript) are in bold.

| | | | | Brier score | | | | |
|---|---|---|---|---|---|---|---|---|
| Model | Uncal. | Platt | Iso Reg | Hist bin | SBA-10 | SBAW-10 | SWC | SWC-HH |
| NB | $0.3670_{0.023}$ | $0.2905_{0.014}$ | $0.2084_{0.009}$ | $0.2350_{0.012}$ | $0.0598_{0.003}$ | $0.0592_{0.003}$ | $0.0522_{0.003}$ | $\mathbf{0.0315}_{0.003}$ |
| DT | $0.2265_{0.008}$ | $0.2200_{0.008}$ | $0.2081_{0.006}$ | $0.2141_{0.007}$ | $0.0595_{0.003}$ | $0.0589_{0.003}$ | $0.0546_{0.004}$ | $\mathbf{0.0354}_{0.003}$ |
| SVM | $0.1034_{0.004}$ | $0.1040_{0.005}$ | $0.1038_{0.005}$ | $0.1070_{0.005}$ | $0.0592_{0.003}$ | $0.0586_{0.003}$ | $0.0549_{0.003}$ | $\mathbf{0.0428}_{0.002}$ |
| RF | $0.1021_{0.002}$ | $0.0524_{0.004}$ | $0.0537_{0.003}$ | $0.0552_{0.003}$ | $0.0597_{0.003}$ | $0.0591_{0.003}$ | $0.0455_{0.003}$ | $\mathbf{0.0344}_{0.003}$ |
| RBFSVM | $0.0756_{0.003}$ | $0.0742_{0.003}$ | $0.0689_{0.003}$ | $0.0706_{0.003}$ | $0.0587_{0.003}$ | $0.0582_{0.003}$ | $0.0442_{0.004}$ | $\mathbf{0.0380}_{0.004}$ |
| GBT | $0.0615_{0.005}$ | $0.0649_{0.005}$ | $0.0593_{0.005}$ | $0.0590_{0.004}$ | $0.0587_{0.003}$ | $0.0581_{0.003}$ | $0.0464_{0.004}$ | $\mathbf{0.0393}_{0.004}$ |
| | | | | Accuracy | | | | |
| Model | Uncal. | Platt | Iso Reg | Hist bin | SBA-10 | SBAW-10 | SWC | SWC-HH |
| NB | $0.8158_{0.011}$ | $0.8160_{0.011}$ | $0.8692_{0.007}$ | $0.8654_{0.007}$ | $0.9632_{0.003}$ | $0.9662_{0.002}$ | $0.9662_{0.003}$ | $\mathbf{0.9816}_{0.001}$ |
| DT | $0.8500_{0.009}$ | $0.8540_{0.008}$ | $0.8568_{0.006}$ | $0.8566_{0.005}$ | $0.9634_{0.003}$ | $0.9656_{0.002}$ | $0.9642_{0.002}$ | $\mathbf{0.9784}_{0.002}$ |
| SVM | $0.9328_{0.004}$ | $0.9340_{0.004}$ | $0.9330_{0.003}$ | $0.9328_{0.003}$ | $0.9638_{0.003}$ | $0.9660_{0.003}$ | $0.9628_{0.003}$ | $\mathbf{0.9744}_{0.002}$ |
| RBFSVM | $0.9454_{0.003}$ | $0.9454_{0.003}$ | $0.9544_{0.003}$ | $0.9524_{0.002}$ | $0.9640_{0.003}$ | $0.9662_{0.002}$ | $0.9694_{0.002}$ | $\mathbf{0.9788}_{0.002}$ |
| GBT | $0.9594_{0.002}$ | $0.9600_{0.003}$ | $0.9590_{0.003}$ | $0.9600_{0.003}$ | $0.9636_{0.003}$ | $0.9660_{0.003}$ | $0.9682_{0.002}$ | $\mathbf{0.9780}_{0.002}$ |
| RF | $0.9634_{0.002}$ | $0.9648_{0.002}$ | $0.9632_{0.002}$ | $0.9634_{0.002}$ | $0.9632_{0.003}$ | $0.9658_{0.003}$ | $0.9700_{0.002}$ | $\mathbf{0.9812}_{0.002}$ |

SWC-HH results were still comparable or better than SBA-10 and the global calibration methods on this data set. In addition, SWC-HH yielded the best accuracy.

The "letter" data set is unusual in that SBAW-10 achieved the best results (Figure 8(e,f) and Table 7, except for the random forest classifier, which is best calibrated using SWC or SWC-HH. SBAW-10 on this data set also outperforms the unweighted SBA-10 approach described by Bella et al. (2009). We interpret this to mean that "letter" exhibits even stronger subpopulation locality than the others we have studied. These results reinforce the importance of employing some form of weighting when calibrating using similarity. However, the choice of 10 neighbors to use does not always work best, and the ideal constant would be difficult to estimate in advance. Therefore, we recommend the use of the entire data set (via SWC or SWC-HH) as a more robust solution.

Table 5: Results for mnist10 ($n_{cal} = 5000$, 10 trials). The best result(s) for each model (within 1 standard error, shown as a subscript) are in bold.

| | | | | Brier score | | | | |
|---|---|---|---|---|---|---|---|---|
| Model | Uncal. | TS | Iso Reg | Hist bin | SBA-10 | SBAW-10 | SWC | SWC-HH |
| NB | $0.8193_{0.015}$ | $0.8031_{0.015}$ | $0.4802_{0.005}$ | $0.5910_{0.013}$ | $0.1729_{0.005}$ | $0.1702_{0.005}$ | $0.1637_{0.004}$ | $\mathbf{0.1164}_{0.004}$ |
| DT | $0.5119_{0.010}$ | $0.5128_{0.008}$ | $0.4832_{0.008}$ | $0.5442_{0.011}$ | $0.1722_{0.005}$ | $0.1695_{0.005}$ | $0.1678_{0.004}$ | $\mathbf{0.1236}_{0.004}$ |
| RF | $0.2977_{0.003}$ | $0.1495_{0.006}$ | $0.1505_{0.004}$ | $0.1620_{0.005}$ | $0.1726_{0.005}$ | $0.1699_{0.005}$ | $\mathbf{0.1323}_{0.005}$ | $\mathbf{0.1311}_{0.005}$ |
| GBT | $0.2199_{0.005}$ | $0.2102_{0.004}$ | $0.2061_{0.004}$ | $0.2090_{0.004}$ | $0.1713_{0.005}$ | $0.1686_{0.005}$ | $0.1469_{0.003}$ | $\mathbf{0.1411}_{0.003}$ |
| SVM | $0.1926_{0.003}$ | $0.1805_{0.003}$ | $0.1786_{0.003}$ | $0.1830_{0.003}$ | $0.1719_{0.005}$ | $0.1692_{0.005}$ | $\mathbf{0.1347}_{0.004}$ | $\mathbf{0.1338}_{0.004}$ |
| RBFSVM | $0.1919_{0.003}$ | $0.1854_{0.003}$ | $0.1766_{0.003}$ | $0.1813_{0.004}$ | $0.1717_{0.005}$ | $0.1690_{0.005}$ | $0.1312_{0.004}$ | $\mathbf{0.1223}_{0.004}$ |
| | | | | Accuracy | | | | |
| Model | Uncal. | TS | Iso Reg | Hist bin | SBA-10 | SBAW-10 | SWC | SWC-HH |
| NB | $0.5895_{0.007}$ | $0.5895_{0.007}$ | $0.6806_{0.004}$ | $0.5779_{0.023}$ | $0.8841_{0.003}$ | $0.8909_{0.003}$ | $0.8991_{0.003}$ | $\mathbf{0.9415}_{0.002}$ |
| DT | $0.6495_{0.008}$ | $0.6495_{0.008}$ | $0.6551_{0.007}$ | $0.6427_{0.007}$ | $0.8847_{0.003}$ | $0.8910_{0.003}$ | $0.8947_{0.004}$ | $\mathbf{0.9368}_{0.002}$ |
| GBT | $0.8589_{0.004}$ | $0.8589_{0.004}$ | $0.8618_{0.003}$ | $0.8631_{0.004}$ | $0.8856_{0.003}$ | $0.8918_{0.003}$ | $0.9018_{0.003}$ | $\mathbf{0.9091}_{0.002}$ |
| RBFSVM | $0.8647_{0.003}$ | $0.8647_{0.003}$ | $0.8777_{0.003}$ | $0.8765_{0.002}$ | $0.8852_{0.003}$ | $0.8914_{0.003}$ | $0.9159_{0.003}$ | $\mathbf{0.9230}_{0.003}$ |
| SVM | $0.8785_{0.002}$ | $0.8785_{0.002}$ | $0.8772_{0.002}$ | $0.8780_{0.003}$ | $0.8853_{0.003}$ | $0.8912_{0.003}$ | $\mathbf{0.9112}_{0.003}$ | $\mathbf{0.9118}_{0.003}$ |
| RF | $0.9006_{0.005}$ | $0.9006_{0.005}$ | $0.9043_{0.004}$ | $0.9023_{0.004}$ | $0.8849_{0.003}$ | $0.8908_{0.003}$ | $\mathbf{0.9123}_{0.003}$ | $\mathbf{0.9134}_{0.003}$ |

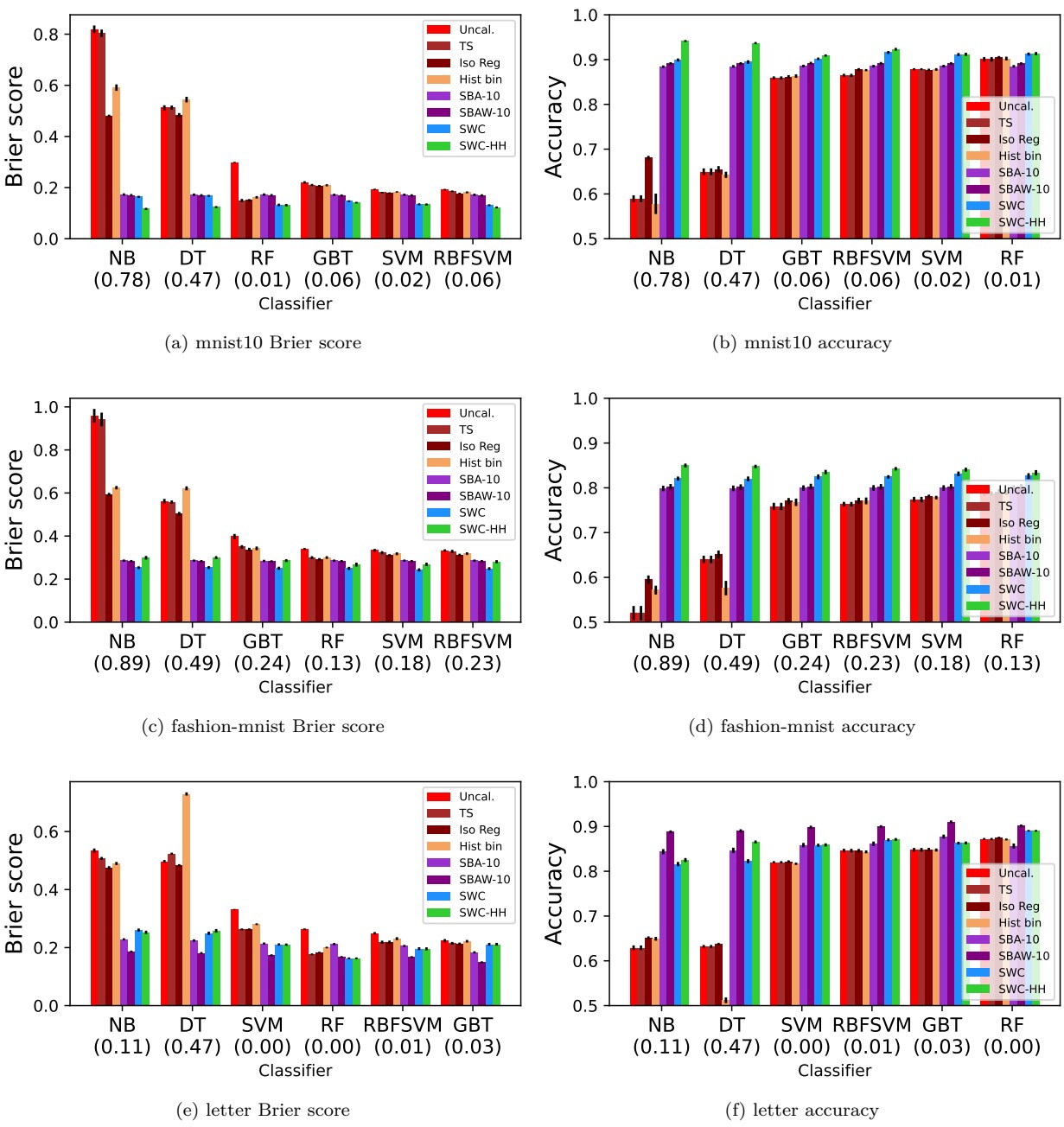

(a) mnist10 Brier score

(b) mnist10 accuracy

(c) fashion-mnist Brier score

(d) fashion-mnist accuracy

(e) letter Brier score

(f) letter accuracy

Figure 8: Calibration performance (left) and accuracy (right) for the multi-class "mnist10", "fashion-mnist", and "letter" data sets (10 trials; error bars indicate one standard error). Classifiers are sorted in order of improvement based on the uncalibrated classifier's score, and average HH values are below each classifier.

Table 6: Results for fashion-mnist ($n_{cal} = 5000$, 10 trials). The best result(s) for each model (within 1 standard error, shown as a subscript) are in bold.

| | Brier score | | | | | | | |
|---|---|---|---|---|---|---|---|---|
| Model | Uncal. | TS | Iso Reg | Hist bin | SBA-10 | SBAW-10 | SWC | SWC-HH |
| NB | $0.9586_{0.032}$ | $0.9415_{0.032}$ | $0.5926_{0.008}$ | $0.6247_{0.008}$ | $0.2857_{0.005}$ | $0.2833_{0.005}$ | $\mathbf{0.2531}_{0.006}$ | $0.2991_{0.008}$ |
| DT | $0.5613_{0.011}$ | $0.5579_{0.007}$ | $0.5032_{0.009}$ | $0.6216_{0.009}$ | $0.2855_{0.005}$ | $0.2832_{0.005}$ | $\mathbf{0.2536}_{0.007}$ | $0.2996_{0.007}$ |
| GBT | $0.3976_{0.012}$ | $0.3493_{0.008}$ | $0.3370_{0.008}$ | $0.3427_{0.009}$ | $0.2845_{0.005}$ | $0.2821_{0.005}$ | $\mathbf{0.2504}_{0.007}$ | $0.2859_{0.006}$ |
| RF | $0.3400_{0.004}$ | $0.2997_{0.007}$ | $0.2913_{0.006}$ | $0.3000_{0.006}$ | $0.2855_{0.005}$ | $0.2832_{0.005}$ | $\mathbf{0.2497}_{0.007}$ | $0.2674_{0.009}$ |
| SVM | $0.3333_{0.007}$ | $0.3226_{0.007}$ | $0.3101_{0.006}$ | $0.3176_{0.007}$ | $0.2856_{0.005}$ | $0.2832_{0.005}$ | $\mathbf{0.2428}_{0.007}$ | $0.2682_{0.008}$ |
| RBFSVM | $0.3327_{0.006}$ | $0.3290_{0.007}$ | $0.3120_{0.006}$ | $0.3182_{0.006}$ | $0.2855_{0.005}$ | $0.2832_{0.005}$ | $\mathbf{0.2486}_{0.007}$ | $0.2807_{0.007}$ |
| | Accuracy | | | | | | | |
| Model | Uncal. | TS | Iso Reg | Hist bin | SBA-10 | SBAW-10 | SWC | SWC-HH |
| NB | $0.5202_{0.016}$ | $0.5202_{0.016}$ | $0.5950_{0.009}$ | $0.5716_{0.010}$ | $0.7986_{0.006}$ | $0.8022_{0.006}$ | $0.8206_{0.005}$ | $\mathbf{0.8498}_{0.004}$ |
| DT | $0.6402_{0.008}$ | $0.6402_{0.008}$ | $0.6510_{0.009}$ | $0.5760_{0.016}$ | $0.7988_{0.006}$ | $0.8014_{0.006}$ | $0.8200_{0.005}$ | $\mathbf{0.8476}_{0.004}$ |
| GBT | $0.7584_{0.008}$ | $0.7584_{0.008}$ | $0.7702_{0.007}$ | $0.7676_{0.008}$ | $0.7996_{0.006}$ | $0.8028_{0.006}$ | $0.8246_{0.005}$ | $\mathbf{0.8352}_{0.005}$ |
| RBFSVM | $0.7634_{0.005}$ | $0.7634_{0.005}$ | $0.7710_{0.007}$ | $0.7708_{0.008}$ | $0.7998_{0.006}$ | $0.8022_{0.006}$ | $0.8244_{0.004}$ | $\mathbf{0.8426}_{0.004}$ |
| SVM | $0.7738_{0.006}$ | $0.7738_{0.006}$ | $0.7804_{0.004}$ | $0.7782_{0.004}$ | $0.7996_{0.006}$ | $0.8022_{0.006}$ | $0.8312_{0.005}$ | $\mathbf{0.8406}_{0.004}$ |
| RF | $0.7872_{0.005}$ | $0.7872_{0.005}$ | $0.7912_{0.006}$ | $0.7902_{0.006}$ | $0.7996_{0.006}$ | $0.8024_{0.006}$ | $0.8262_{0.006}$ | $\mathbf{0.8336}_{0.006}$ |

Table 7: Results for letter ($n_{cal} = 5000$, 10 trials). The best result(s) for each model (within 1 standard error, shown as a subscript) are in bold.

| | Brier score | | | | | | | |
|---|---|---|---|---|---|---|---|---|
| Model | Uncal. | TS | Iso Reg | Hist bin | SBA-10 | SBAW-10 | SWC | SWC-HH |
| NB | $0.5342_{0.007}$ | $0.5073_{0.005}$ | $0.4753_{0.005}$ | $0.4899_{0.005}$ | $0.2284_{0.004}$ | $\mathbf{0.1863}_{0.003}$ | $0.2603_{0.005}$ | $0.2527_{0.006}$ |
| DT | $0.4975_{0.004}$ | $0.5237_{0.003}$ | $0.4839_{0.004}$ | $0.7296_{0.006}$ | $0.2243_{0.004}$ | $\mathbf{0.1812}_{0.003}$ | $0.2492_{0.006}$ | $0.2576_{0.006}$ |
| SVM | $0.3311_{0.002}$ | $0.2629_{0.003}$ | $0.2632_{0.003}$ | $0.2806_{0.003}$ | $0.2141_{0.003}$ | $\mathbf{0.1737}_{0.003}$ | $0.2107_{0.004}$ | $0.2102_{0.004}$ |
| RF | $0.2633_{0.003}$ | $0.1769_{0.003}$ | $0.1830_{0.003}$ | $0.2002_{0.003}$ | $0.2121_{0.004}$ | $0.1679_{0.003}$ | $\mathbf{0.1630}_{0.003}$ | $\mathbf{0.1630}_{0.003}$ |
| RBFSVM | $0.2495_{0.004}$ | $0.2186_{0.005}$ | $0.2189_{0.005}$ | $0.2308_{0.005}$ | $0.2056_{0.004}$ | $\mathbf{0.1673}_{0.003}$ | $0.1963_{0.005}$ | $0.1957_{0.005}$ |
| GBT | $0.2237_{0.006}$ | $0.2147_{0.005}$ | $0.2129_{0.004}$ | $0.2219_{0.004}$ | $0.1835_{0.004}$ | $\mathbf{0.1491}_{0.003}$ | $0.2114_{0.005}$ | $0.2115_{0.005}$ |
| | Accuracy | | | | | | | |
| Model | Uncal. | TS | Iso Reg | Hist bin | SBA-10 | SBAW-10 | SWC | SWC-HH |
| NB | $0.6291_{0.005}$ | $0.6291_{0.005}$ | $0.6511_{0.003}$ | $0.6492_{0.004}$ | $0.8438_{0.006}$ | $\mathbf{0.8881}_{0.003}$ | $0.8160_{0.005}$ | $0.8251_{0.004}$ |
| DT | $0.6321_{0.003}$ | $0.6321_{0.003}$ | $0.6375_{0.003}$ | $0.5119_{0.006}$ | $0.8465_{0.006}$ | $\mathbf{0.8904}_{0.003}$ | $0.8223_{0.005}$ | $0.8654_{0.003}$ |
| SVM | $0.8202_{0.002}$ | $0.8202_{0.002}$ | $0.8217_{0.003}$ | $0.8168_{0.003}$ | $0.8584_{0.005}$ | $\mathbf{0.8982}_{0.004}$ | $0.8581_{0.003}$ | $0.8589_{0.003}$ |
| RBFSVM | $0.8461_{0.004}$ | $0.8461_{0.004}$ | $0.8459_{0.004}$ | $0.8434_{0.003}$ | $0.8615_{0.005}$ | $\mathbf{0.8998}_{0.003}$ | $0.8701_{0.003}$ | $0.8711_{0.003}$ |
| GBT | $0.8479_{0.004}$ | $0.8479_{0.004}$ | $0.8492_{0.003}$ | $0.8474_{0.003}$ | $0.8774_{0.004}$ | $\mathbf{0.9103}_{0.003}$ | $0.8629_{0.003}$ | $0.8633_{0.003}$ |
| RF | $0.8722_{0.002}$ | $0.8722_{0.002}$ | $0.8746_{0.002}$ | $0.8710_{0.002}$ | $0.8565_{0.006}$ | $\mathbf{0.9013}_{0.003}$ | $0.8904_{0.002}$ | $0.8904_{0.002}$ |

