# OpenReview forum: "Hidden Heterogeneity: When to Choose Similarity-Based Calibration"
_TMLR — Accepted by TMLR_

### Review · Reviewer_94q8 · 2022-10-19

**Summary Of Contributions:**

The authors propose a notion called 'hidden heterogeneity', which describes the situation when subpopulations having the same predicted probability exhibit different true conditional probabilities. They propose to remedy the situation with a local regression correction that can improve both the probability prediction and classification accuracies. They perform fairly extensive empirical evaluations on several datasets, and their method has the best improvements when the model underfits the data.


**Audience:**

Yes

**Broader Impact Concerns:**

- There are no strong ethical implication from this piece of work on probability calibrations.


**Claims And Evidence:**

Yes

**Requested Changes:**

- Discuss the relation of the proposed method to model mis-specification.
- Discuss the rationale behind using Hellinger distance and RFprox as similarity measures in greater details.
- Some of the texts in Figures 1, 2, 6 are too small to be readable. Please fix.


**Strengths And Weaknesses:**

Strengths:
- The proposed algorithm is fairly simple to understand. Instead of calibrating the probabilities globally, the main idea is to calibrate the probabilities locally using the features x too when there are model mis-specification. The design and steps in the algorithm are very clearly written.

- The authors have done a fairly extensive evaluations on a few standard ML datasets such as MNIST and letters, and also on larger image datasets such as CIFAR10 and CIFAR100. They show improved probability calibrations compared to global calibration methods and competing local calibration method such as SBA-10, especially for weaker classifiers.

- In the paper there are fairly interesting empirical analysis on how the local calibration improves classification accuracy (section 5.3), and how the calibration support set helps discover anomalies in the prediction (section 5.5).


Weaknesses:
- The current paper does not discuss the relation between the proposed method to model mis-specification. When different subpopulations U_i, U_j having the predicted probabilities f(x) but different true conditional probabilities, isn't that a sign of model mis-specification? (consider the cats vs birds example in section 3). When such a situation arise, instead of calibrating the probabilities with local regression (a post-hoc fix), we could revise the model using local regression or mixture of experts type of model. The authors do not seem to consider such a possibility but go straight to correcting the model using local probability calibrations.

- Certain choices in the scoring of hidden heterogeneity and the algorithm seem arbitrary and not well-motivated enough. How different would the results be if we use a KL-divergence or L1-distance measure on the probability vectors instead of Hellinger distance? The algorithm also makes use of random forests in its computation of similarities between items, a decision that the authors did not discuss their reasoning. What would happen if we replace the RFprox measure with some other measures?

- In Algorithm 1, the local probability predictor g_t works with a potentially much smaller sample. Wouldn't this cause instability problems in the calculation of the HH measure? For example, in Figure 1c, when the base classifier is powerful enough (random forest), local calibration methods such as SBA-10 or the proposed method actually hurts probability calibration for small sample size compared to global method such as Platt scaling, when the number of calibration items are small. Also, how is the radius r for neighborhood chosen?

---

> ### Author Response · Authors · 2022-11-11
> **Review response**
>
> Thank you for the careful and thorough reading of this paper.  We have addressed the requested changes in an update to the paper (highlighted in blue) as follows:
>
> 1. Model mis-specification:  This can be one source of hidden heterogeneity (HH), among others mentioned on page 4 (section 3), and we have added it to the list with additional discussion.  The reason we do not propose to try to amend or revise the model is that we are addressing the case in which the model is provided as a fixed (possibly proprietary) component, and we cannot assume access to (or any knowledge of) the original training data (see Section 1, paragraph 2 and Section 2, paragraph 1).  Our focus is on what can be done to ensure that the model’s predictions are well calibrated when applied to our own data set (post-hoc calibration), a common strategy when the original training data is not available.
>
> 2. The reviewer makes a very good observation that the choice of similarity measure employed by SWC is open to the investigator.  We updated section 4, paragraph 1 to  emphasize that one could make a different choice based on knowledge of the domain (e.g., weighting particular features in custom ways).  Since the experiments in this paper use benchmark data sets (no domain knowledge), we chose to use a standard similarity measure (RFProx) that could be used in all cases and is flexible enough to develop a problem-specific feature weighting guided by labeled data.  It is important that the similarity measure pay strong attention to the predicted probabilities from the original classifier.  Because they are supervised, similarity measures such as RFProx can learn that the predicted probabilities have strong predictive power.  If unsupervised similarity measures are employed, additional mechanisms will be needed to ensure that they give high importance to the predicted probabilities.  We added this discussion and justification to section 4 as well.
>
> Another implementation choice is the use of Hellinger distance to measure distance in the probability simplex when computing HH (section 3, equation 1).  As noted in the paper, but perhaps not clearly enough, Hellinger distance is the equivalent of Euclidean or L2 distance for probabilities and commonly used for this purpose.  Because KL-divergence is asymmetric, it would not serve our requirement for a metric.   We updated the text in Section 3 to clarify.  Just as when choosing a distance metric for feature vectors, the investigator could choose a different distance metric informed by domain-specific knowledge.  We added this note to section 3.
>
> 3. Stability of HH calculation and how to choose probability radius r:  To obtain good (stable) HH estimates, we recommend selecting a radius r that ensures the set g(t) trains on is not excessively small.  In our experiments, r=0.1 worked well for the tabular data sets, while the much larger ImageNet data sets worked well with r=0.05 (section 5.4, paragraph 1).  We added this explicit advice about the parameter setting to section 3.  Relatedly, if the calibration set as a whole is too small then a similarity-based method may not have enough information to perform good calibration.  We previously made this point in the Conclusions and have now also added a statement about it to section 5.2.
>
> 4. We increased the font sizes in Figures 1, 2, and 6.
>
> Thank you again for your suggestions and improvements.

---

### Review · Reviewer_ijrk · 2022-10-23

**Summary Of Contributions:**

The paper challenges the convention that samples with the same predicted probability should be given the same calibration correction, and proposes to detect subpopulations where better calibration could improve accuracy, which the paper calls "hidden heterogeneity". The paper introduces two "local" calibration methods, Similarity-Weighted Calibration (SWC) and SWC-HH that leverage similarity between samples in terms of features. SWC and SWC-HH differ slightly, in that SWC weighs all points in a calibration dataset based on similarity to the given point x that is being studied, while SWC-HH uses only a subset of these points. The paper tests the method on tabular and image datasets.


**Audience:**

Yes

**Claims And Evidence:**

No

**Requested Changes:**

- [Critical in my opinion] Experiments where model is retrained, or more powerful model classes used, and show that they don’t outperform local calibration methods.
- [Critical in my opinion] Experiments where hidden heterogeneity is introduced using different types of data mechanisms and showing how the calibration methods perform in this case.
- [Critical in my opinion] Since SBA-10 is a key baseline compared against in this paper, it would be good to confirm if SBA-10 uses Euclidean distance , cf. "Our implementation of SBA-10 employs Euclidean distance to identify the nearest neighbors, which we believe to be the method employed by Bella et al.".
- [Strengthen the work] I found the results in Section 5.4 a bit confusing. It sounds like the similarity measure used in SWC and SWC-HH is a feature space learned by the neural net. But that is followed by this line “We again used a learned RF proximity function to compute similarity.”. Clarification would be great.
- [Strengthen the work] If the local calibration methods are not tied to a specific similarity measure, like random forest proximity, this could be made clearer in Section 4 when introducing the methods.
- [Strengthen the work] For SWC-HH, from this line in Section 4: “HH, which is computed separately for each test item t, is provided as an additional input for calibration”, on which dataset split is this HH computed on?

**Strengths And Weaknesses:**

Strengths:
- A fresh take on a classic problem – calibration – and challenging the convention that samples with the same predicted probability should be given the same calibration correction.
- Experiments on different types of datasets, not just data from one type of modality.

Weaknesses:
- Hidden heterogeneity, the problem that the paper defined and is trying to solve for, is defined by comparing predictions from the original model and a newly trained model (a random forest, according to page 3). Hidden heterogeneity is then used as a gold standard to compare calibration methods. Yet, the calibration methods introduced in this paper leverage random forest proximity. This feels a bit like “training for the test”. This would be more convincing if there were some synthetic data experiments where hidden heterogeneity is introduced using different types of data mechanisms and showing that the calibration methods can still perform well. Or using different types of distances in the calibration methods – essentially a sensitivity analysis of distance type and how that impacts results.
- Could go deeper to try to explain some of the observed phenomenon. For example, in the results in Section 5.3, the random forest classifier “somewhat to our surprise, exhibits more areas with hidden heterogeneity”. Do the authors have any ideas why? In Section 5.4, when the authors said “using random forests to measure hidden heterogeneity in neural network latent spaces may fail to detect HH even though a better representation (or a better classifier) could detect that it is present”, again, a few sentences to discuss why would improve the presentation of these ideas and help the reader gain intuition into failure modes of the method.
- Even if hidden heterogeneity (subpopulations where better calibration could improve accuracy) does exist, it is not clear that improving the calibration for this subpopulation is the preferred method to correct for this, if a better classifier could be trained. A very personalized calibration function could be harder to deploy in practice than a global calibration function. If a better model could be trained such that these subpopulations have better accuracy, this local calibration idea might not be needed. I think adding some of these baselines could be interesting: 1/ retraining the model, upsampling subpopulations found to have hidden heterogeneity; 2/ using more powerful model classes.

---

> ### Author Response · Authors · 2022-11-11
> **Review response**
>
> Thank you for the careful and thorough reading of this paper.  We welcome the opportunity to improve the paper and respond to the requested changes as follows.  All changes to the paper are highlighted in blue.
>
> > *“training for the test” … [apparent use of random forests in HH classifier and SWC similarity measure]*
>
> To clarify, random forests are not used in both operations.  The g(t) classifier employed in HH calculation is a bagged ensemble of carefully pruned decision trees, and it is not a random forest.  (This helps explain why we do not observe an HH of zero for random forest classifiers.)  We amended Section 5.4 which incorrectly referred to the HH calculation as using random forests.
>
> The random forest used by RFprox defines a real-valued similarity measure; it is not used as a classifier.  We updated section 4 to emphasize that one could make other choices for the similarity measure (in fact, a domain-specific one would likely perform even better) and to discuss the importance of using a supervised measure so that the predicted probabilities influence the learned concept of locality.
>
> > *Could go deeper to try to explain some of the observed phenomenon … Section 5.3, 5.4.*
>
> We added more explanations to Sections 5.3 and 5.4, as suggested.
>
> > *Experiments where model is retrained, or more powerful model classes used, and show that they don’t outperform local calibration methods.*
>
> This is a good strategy when access to the original training data is available.  However, we are addressing the case in which the model is provided as a fixed (possibly proprietary) component, and we cannot assume access to (or any knowledge of) the original training data (see Section 1, paragraph 2 and Section 2, paragraph 1).  Our focus is on what can be done to ensure that the model’s predictions are well calibrated when applied to our own data set (post-hoc calibration), a common strategy when the original training data is not available.
>
> > *Experiments where hidden heterogeneity is introduced using different types of data mechanisms and showing how the calibration methods perform in this case.*
>
> There are three factors that influence HH: the model (classifier), the data distribution, and the data representation. This paper conducts experiments that explore the first and second factors: how different model choices influence HH and calibration performance (Sections 5.2, 5.3, 5.4) and how changes in the data distribution impact calibration (Section 5.5); the latter is most relevant to the reviewer’s suggestion.  Alternate representations and their impact on HH and calibration are not explored in the current paper.  We have added this as a useful direction for future work (Section 6).
>
> > *Since SBA-10 is a key baseline compared against in this paper, it would be good to confirm if SBA-10 uses Euclidean distance*
>
> Thanks for making this point.  The distance measure is not specified in Bella et al.’s 2009 paper.  In response to your review, we contacted the authors and they confirmed that Euclidean distance was used, so we have removed this speculative comment.  In addition, they have subsequently employed inverse distance weighting when combining the nearest neighbors (previously listed as a future work direction in the 2009 paper), so we have conducted additional experiments with this variation on SBA and added them to Appendix A (denoted as SBAW).  We can view SBAW as an intermediate choice between SBA and SWC, as it adopts the distance weighting of SWC but not the other aspects (supervised distance metric, HH filtering) that make SWC-HH the strongest method overall.
>
> > *I found the results in Section 5.4 a bit confusing. It sounds like the similarity measure used in SWC and SWC-HH is a feature space learned by the neural net. But that is followed by this line “We again used a learned RF proximity function to compute similarity.”. Clarification would be great.*
>
> That is correct: SWC and SWC-HH use an RFprox similarity measure that is trained in the feature space learned by the neural network.  We updated the wording in Section 5.4 to clarify this.
>
> > *If the local calibration methods are not tied to a specific similarity measure, like random forest proximity, this could be made clearer in Section 4 when introducing the methods.*
>
> We updated the text in Section 4 to clarify that the investigator can choose the similarity measure as well as clarifying why we chose to use RFprox, to develop a problem-specific feature weighting guided by labeled data.
>
> > *For SWC-HH, from this line in Section 4: “HH, which is computed separately for each test item t, is provided as an additional input for calibration”, on which dataset split is this HH computed on?*
>
> HH is computed for each test item using the relevant neighborhood of the calibration set (see Algorithm 1, step 1).  We added a reference back to Algorithm 1 at this point in the text to aid the reader.
>
> Thank you again for these helpful suggestions.

---

### Review · Reviewer_DmA5 · 2022-10-29

**Summary Of Contributions:**

The paper defines the new notion of hidden heterogeneity (HH) about situations where a model is predicting the same class probability vector to instances for which the true posterior probabilities are not the same. It then defines local calibration methods SWC and SWC-HH to reduce HH and improve classifier performance as measured by Brier score, and by accuracy as well. Experiments are performed on multiple tabular and image datasets. Results show that the proposed methods outperform Platt scaling, histogram binning, and temperature scaling in many situations.


**Audience:**

Yes

**Broader Impact Concerns:**

No further concerns beyond what was mentioned above.


**Claims And Evidence:**

No

**Requested Changes:**

The experiments should include some state-of-the-art calibration methods and also the classical isotonic calibration. The paper by Luo et al 2021 should be discussed in more detail and the relationship of HH to Luo's work should be discussed. It would be discussed more when the proposed methods are useful and when they would not help because of low amount of HH. The definition of calibration should be fixed.

**Strengths And Weaknesses:**

Strengths:
* The text is clear and understandable.
* There are experiments about tabular data as well as images.

Weaknesses:
* Local calibration has been studied before by Luo et al 2021 whom the current paper cites briefly but not in sufficient detail, given the relevance to this work. Therefore, the concept of hidden heterogeneity is not fully new, although it is new as a term. In some sense, HH means just 'not locally calibrated'.

* Following this, it would be essential that the paper would compare against the method proposed by Luo et al 2021. Furthermore, there are many more methods that should be considered to include in the comparisons. Currently, the paper only compares against the most basic calibration methods. For example, the paper states about Platt scaling that 'No further improvements were achieved beyond 500 calibration items'. Of course, this is very much expected because Platt scaling only needs to fit 2 parameters which is easy to do on a small dataset and more instances do not help to improve much anymore. Therefore, it would be important to include some other existing calibration methods that either have more parameters or that are non-parametric. For example, including isotonic calibration in the experiments would be absolutely essential.

* There is enough HH that can be exploited and improved over using local modelling only when the original classifier was not doing it already. As the authors discovered, neural networks on images are already exploiting local information and hence, HH values were too low to be exploited further. This highlights the general limitation of the proposed methods - they are only useful if the original classifier was not very good.

* The definition of calibration at the beginning of the introduction is wrong. It has been defined through P(Y|X), whereas actually the conditioning should not be on X but on the output of the classifier. Otherwise, only the Bayes-optimal model would be well-calibrated. However, according to the standard definition of calibration, for example even a constant classifier that predicts the true class distribution P(Y) is also calibrated.

* 'Our implementation of Platt scaling ...' sounds like the authors would have done something non-standard. However, this is exactly as how J. Platt originally proposed it and even the Python scikit-learn implementation of Platt scaling is doing the same, i.e. the targets are not {0,1} but instead smoothed probabilities.

---

> ### Author Response · Authors · 2022-11-11
> **Review response**
>
> Thank you for the careful and thorough reading of this paper.  Your suggestions inspired several improvements and additional experiments that are now included in the paper.  All changes to the paper are highlighted in blue.
>
> > *The experiments should include some state-of-the-art calibration methods and also the classical isotonic calibration.*
>
> Including Isotonic Regression (IR) is an excellent suggestion since IR, Platt scaling, and histogram binning are the most commonly used calibration methods.  We have added IR to the suite of state-of-the-art methods evaluated in all of our experiments.  We found that it often improves a bit over Platt/temperature scaling and histogram binning, but it still falls short of similarity-based calibration methods (see updated Figures 1, 7, and 8), except on the image classifier experiments, where it is approximately equivalent to temperature scaling (see updated Figure 4).
>
> > *The paper by Luo et al 2021 should be discussed in more detail and the relationship of HH to Luo's work should be discussed.*
>
> As far as we can determine, the work by Luo et al. has not yet been published in a peer-reviewed venue, and no code is available.  It was submitted to ICLR 2022, but not accepted, so it may not be in its final form (https://openreview.net/forum?id=T_p2GaXuGeA),.  We have expanded our discussion of their work in Section 2 based on the latest arXiv version. We would expect the LoRe approach to perform worse than our method for several reasons. First, LoRe limits the computation of the local neighborhood to items that fall into the same probability bin for the probability of the top-predicted class. Our method includes the entire predicted distribution in the augmented feature space, which enables local calibration to consider finer-grained probability similarities. Second, LoRe’s bin-based neighborhood could result in very sparse neighborhoods for some test points (when the bin contains few calibration points), and this could lead to high-variance calibrated probability estimates. Another drawback of LoRe is that it only calibrates the probability of the highest-probability class label. While this suffices for prediction, it is not sufficient for other tasks such as computing expected costs of misclassification (in cost-sensitive problems) or re-estimating class probabilities (as in the work of Alexandari, et al. (arxiv 1901.06852; “EM with Bias-Corrected Calibration is Hard-To-Beat at Label Shift Adaptation,” ICML 2020). This discussion is now included in Section 2.
>
> > *It would be discussed more when the proposed methods are useful and when they would not help because of low amount of HH. [...] This highlights the general limitation of the proposed methods - they are only useful if the original classifier was not very good.*
>
> We agree with the first statement, and the paper makes the point that similarity-based calibration provides an advantage only in the presence of HH in Section 5.4 and Section 6.  However, it is not the case that these methods are “only useful if the original classifier was not very good.”  The original classifier may have been a very high performer on the original evaluation data set, but data shift is present in the new test set.  Alternatively, it may have been a good model that made tradeoffs for global regularization that led to under-fitting in some local regions.  We expanded the discussion of causes of HH in Section 3.
>
> There are also benefits provided by similarity-based calibration even if Brier score itself is not improved.   Similarity-based calibration can detect data shift that will impact calibration and alert the investigator when it is present (see Section 5.5).  Further, this approach can identify individual test items that lack sufficient support in the calibration set even if the data set average HH is low (see Section 5.5).
>
> > *The definition of calibration should be fixed.   It has been defined through P(Y|X), whereas actually the conditioning should not be on X but on the output of the classifier.*
>
> We have added the notation f(X) to refer to the output of the classifier, which is distinguished from the true class probability distribution for item X, notated as P(Y|X).  Thank you for helping us improve this definition.
>
> > *'Our implementation of Platt scaling ...' sounds like the authors would have done something non-standard. However, this is exactly as how J. Platt originally proposed it and even the Python scikit-learn implementation of Platt scaling is doing the same, i.e. the targets are not {0,1} but instead smoothed probabilities.*
>
> We have removed “Our implementation of Platt scaling” to clarify that this is the standard method.
>
> Thanks again for these helpful suggestions.

---

### Decision · Action_Editors · 2022-12-03

**Recommendation:** Accept with minor revision

**Comment:**

The paper studies the "hidden heterogeneity" (HH) problem, wherein global calibration of classifier outputs may fail due to the presence of latent sub-populations with similar predictions, but different characteristics. The paper proposes a metric to quantify HH, and proposes local calibration methods to combat it.

### Reviewer recommendations

Classifier calibration is a fundamental problem, and progress on this topic is of broad interest. Reviewers found the paper's treatment of calibration to be interesting, with results presented on both tabular and image data.

From the reviewers' final recommendations, two were (leaning) positive, while another leaned negative. The main critiques raised by the reviewers were as follows.

(1) _Applicability to powerful models_. One concern was whether the approach is limited to cases where the base model performs poorly. The authors responded that they focus on settings where the base model is given, and there is no access to the training data. They also noted that strong models could suffer from HH under distribution shift.

(2) _Impact of model misspecification_. One question was whether HH was simply a symptom of misspecification of the base model, which might be fixed by constructing an MoE-style model. The response was similar to the above.

(3) _Impact of different mechanisms of HH_. One question was whether different causes for HH could result in varied performance. The author's response was that they evaluate the impact of the model and data distribution, with the impact of data representation left for future work.

(4) _Limitation to post-hoc calibration_. One critique was that the paper focusses on post-hoc calibration, which may be a niche setting.

(5) _Comparison to existing work_. The preprint of Luo et al., which also studies local calibration, was not discussed in detail or compared to in the original version. The author response included a more detailed discussion, including limitations of this work.

(6) _Risk of fitting on small sample_. The algorithm relies on local probability estimates fit on a potentially small sample, which could bias the HH score. The author responded that this might indicate scenarios where good similarity-based calibration is not possible.

(7) _Justification for design choices_. The paper makes some design decisions (e.g., the use of Hellinger distance, RFProx as a similarity measure) that are not ablated. The response noted that the Hellinger distance is equivalent to the L2 distance between probabilities, and that RFProx is a standard similarity measure.

Of these, I find the author response (or paper's treatment) to be adequate for (1)-(5), and the reviewers mostly concur. I find (6) and (7) to potentially require more discussion, even after the revision. The discussion of the choice of RFProx in particular could be treated in more detail, and contrast against more common measures (e.g., Gaussian kernel similarity). That said, the paper does demonstrate that the use of RFProx leads to reasonable results in many practical settings.


### AE recommendation

The TMLR guidelines for acceptance are based on the following two questions:

(a) _Are the claims made in the submission supported by accurate, convincing and clear evidence?_

(b) _Would some individuals in TMLR's audience be interested in the findings of this paper?_

For (a), most of the critiques above were adequately addressed in the response. There are some remaining points that would improve the clarity of the work. Nonetheless, in my estimation, the paper's claims are generally well-supported.

For (b), the general reviewer consensus is the paper provides an interesting perspective on calibration. This is supported by my reading. I thus believe some readers would find the results of interest.

Given these, we recommend acceptance with revisions as suggested below.


### Suggested revisions

- Amend the definition of calibration. It would be easiest to provide a precise mathematical definition. If a textual description is to be retained, then it should be reworded to be clearer.

- Section 3 could be split into two sub-sections, one for "What is HH?", and another for "How do we detect HH?"

- (optional) For Algorithm 1 and 2, perhaps x_t rather than t is clearer to refer to the test item.

- Equation 3, 4, 5: use \text{.} for "Brier", "sim"

- In Section 4, what does the notation \langle x_t, \hat{p}_t \rangle mean? Vector concatenation? If so, perhaps [ x_t, \hat{p}_t ] or similar would be clearer (as the former is sometimes used to denote inner product).

- Regarding the use of RFProx in Section 4:

	- It is stated that "If a domain-specific similarity measure is not available, one can use a supervised similarity measure that can learn to place high importance on the predicted probabilities. We employ such a measure: the random forest proximity function (RFprox)." It is not clear from the description why the RFProx necessarily learns to place high importance on the predicted probabilities. Is the claim that it _can_ learn that these probabilities are _typically_ very important? Clarification is suggested.

	- Further to the above, it would be good to make explicit what features are fed into the RFProx method. Is it the concatenation of both x_t and \hat{p}_t? If the stated goal is to place high importance on the latter, why not just use the latter? Further discussion is suggested.

	- It is mentioned that RFProx can be viewed as a kernel. The reader may naturally wonder whether one can use more standard kernels, such as the linear or Gaussian kernel, on top of the concatenation of x_t and \hat{p}_t. This would appear place some importance on the predicted probabilities to measure calibration. Are such choices admissible? What is the reason for choosing RFProx over these? My understanding from the response is that these kernels would be considered "unsupervised", and thus potentially not adequately balance similarities in feature and predicted probability space; this point is however not too clear from the current Section 4. Further to the above, it is not clear whether just constructing a kernel over the predicted probabilities would not then be admissible. More discussion is suggested.

	- If indeed other standard kernels are admissible, it would be ideal if some additional experiments could be presented (potentially on a few setups) with one of these choices. This could demonstrate how important it is to pick an supervised similarity measure.

- The author response to a few critiques relied on the focus being on post-hoc settings. This is reasonable. It might however be interesting to discuss what strategies might be appropriate in settings where one does have the ability to modify the original model. Is it the case here that simply increasing model capacity fixes HH? Would there still be value in similarity-based local calibration?

- The authors stated that "Relatedly, if the calibration set as a whole is too small then a similarity-based method may not have enough information to perform good calibration. We previously made this point in the Conclusions and have now also added a statement about it to section 5.2". From my reading, these points were not too clear. They could be made more explicit.


**Audience:**

The general reviewer consensus is the paper provides a fresh perspective on calibration, which some readers would find interesting. This is supported by my reading.

**Claims And Evidence:**

Reviewers had a number of questions based on the initial submission. The revision adequately addressed many of these. There are a few remaining points that should be addressed in the final revision so as to make the paper's claims more clear and convincing. Overall, however, I find the paper's claims generally well-supported.

---

> ### Author Response · Authors · 2022-12-27
> **Response to final suggested revisions (part 1)**
>
> Thank you for the very clear summary of the paper and the reviews.  We appreciate all of the suggestions for further improvement of the paper.  We uploaded a new version that highlights in blue all changes with respect to the previous version.
>
> > _Amend the definition of calibration. It would be easiest to provide a precise mathematical definition. If a textual description is to be retained, then it should be reworded to be clearer._
>
> We replaced the current textual description (first paragraph of Section 1) with a more precise mathematical definition.
>
> > _Section 3 could be split into two sub-sections, one for "What is HH?", and another for "How do we detect HH?"_
>
> Done.
>
> > _(optional) For Algorithm 1 and 2, perhaps x_t rather than t is clearer to refer to the test item._
>
> Done.
>
> > _Equation 3, 4, 5: use \text{.} for "Brier", "sim"_
>
> Thank you, we have employed \text{.} as recommended.
>
> > *In Section 4, what does the notation \langle x_t, \hat{p}_t \rangle mean? Vector concatenation? If so, perhaps [ x_t, \hat{p}_t ] or similar would be clearer (as the former is sometimes used to denote inner product).*
>
> Good point. Yes, we intended vector concatenation.  Following your suggestion, we replaced \langle/\rangle with [ ] and explained the notation employed (Section 4, paragraph 1).
>
> > *Regarding the use of RFProx in Section 4:
> > It is stated that "If a domain-specific similarity measure is not available, one can use a supervised similarity measure that can learn to place high importance on the predicted probabilities. We employ such a measure: the random forest proximity function (RFprox)." It is not clear from the description why the RFProx necessarily learns to place high importance on the predicted probabilities. Is the claim that it can learn that these probabilities are typically very important? Clarification is suggested.*
>
> We have added clarification as suggested, which is in agreement with the editor’s understanding.
>
> > *Further to the above, it would be good to make explicit what features are fed into the RFProx method. Is it the concatenation of both x_t and \hat{p}_t? If the stated goal is to place high importance on the latter, why not just use the latter? Further discussion is suggested.*
>
> RFProx adheres to the definition of all sim() measures used by SWC (Section 4, paragraph 1), so it operates on the augmented feature space, which is the concatenation of x_t and \hat{p}_t.  This may be more clear with the text added as noted above.  The hypothesis behind this work is that both sources of information are of value.  Using only \hat{p}_t to determine calibration is what most existing methods (e.g., Platt scaling, histogram binning, isotonic regression) do.  Because HH is (by definition) “hidden” within \hat{p}, methods that use only \hat{p}  cannot address it, as shown in the experimental results.  Likewise, a “calibration” method that only examines x_t would simply replace the previous classifier with a new classifier.  Using both x_t and \hat{p}_t enables the correction (calibration) of \hat{p}_t to be informed by, but not entirely superseded by, x_t.
>
> We have updated the discussion of RFProx in Section 4, paragraph 1, to make this more clear in the paper.
>
> > *It is mentioned that RFProx can be viewed as a kernel. The reader may naturally wonder whether one can use more standard kernels, such as the linear or Gaussian kernel, on top of the concatenation of x_t and \hat{p}_t. This would appear place some importance on the predicted probabilities to measure calibration. Are such choices admissible? What is the reason for choosing RFProx over these? My understanding from the response is that these kernels would be considered "unsupervised", and thus potentially not adequately balance similarities in feature and predicted probability space; this point is however not too clear from the current Section 4. Further to the above, it is not clear whether just constructing a kernel over the predicted probabilities would not then be admissible. More discussion is suggested.*
>
> We have added text to clarify why it is important for the sim() function to determine how much to weight the \hat{p}_t versus the x_t features. A standard kernel would fix the relative weight of these two components rather than learning it. This is why we use supervision (informed by the labels). Another advantage of RFProx is that it can apply different weightings in different parts of the [X,P] space.
>
> (please see next comment for continuation)

---

> ### Author Response · Authors · 2022-12-27
> **Response to final suggested revisions (part 2)**
>
> (continued)
>
> > *If indeed other standard kernels are admissible, it would be ideal if some additional experiments could be presented (potentially on a few setups) with one of these choices. This could demonstrate how important it is to pick an supervised similarity measure.*
>
> We know of no standard kernels that would work (since they would by default treat each feature of the augmented space equally). We added a paragraph discussing why and proposing a possible line of research (using multiple kernel learning) to combine a kernel over the x_i space with a kernel over the p_i space. This would still not be as flexible as RFProx.
>
> > *The author response to a few critiques relied on the focus being on post-hoc settings. This is reasonable. It might however be interesting to discuss what strategies might be appropriate in settings where one does have the ability to modify the original model. Is it the case here that simply increasing model capacity fixes HH? Would there still be value in similarity-based local calibration?*
>
> As noted in Section 3.1, there are many possible causes of HH.  If limited model capacity is the cause, and one has access to the original training data, then retraining with a more expressive model could possibly reduce HH, but there is the associated risk of overfitting to consider.  A less risky approach could be to use HH as a fine-grained diagnostic (finer than just looking at overall loss) to pinpoint regions in feature space that could benefit from collecting additional training data or training a local expert model to override or contribute to the global model.  However, other causes of HH such as data shift cannot be addressed by increasing model capacity and re-training on the original data set, so similarity-based calibration still has a role even when the original training data is available.  Because the paper focuses on post-hoc calibration problems, there doesn’t seem to be a natural place in this paper to discuss variations when re-training is an option, so we prefer to postpone this discussion to a larger conversation in follow-on work.
>
> > *The authors stated that "Relatedly, if the calibration set as a whole is too small then a similarity-based method may not have enough information to perform good calibration. We previously made this point in the Conclusions and have now also added a statement about it to section 5.2". From my reading, these points were not too clear. They could be made more explicit.*
>
> Section 5.2 states that “Similarity-based calibration (SBA-10, SWC, and SWC-HH) generally did not perform well with small calibration sets” while discussing Figure 1, which also makes this point visually.  Section 5.3 now states “When HH is low or there is not much calibration data, global methods such as Platt scaling are recommended.”  We also added more explanation for why local methods may need more calibration data than global methods (they have more degrees of freedom) to the first paragraph of Section 6.
>
> In addition to making the requested revisions, we also made several small wording clarifications and de-anonymized the paper, including author contributions (at the end) and a link to all source code and scripts (first paragraph of Section 5).  Thank you again for your guidance in this process!